# Diverse and Sparse Mixture-of-Experts for Causal Subgraph–Based Out-of-Distribution Graph Learning

**Jerry Sun & Mohamed A. Hassan & Yaoyu Zhang & Wanying Zhang & Chi-Guhn Lee**
Department of Mechanical and Industrial Engineering
University of Toronto
Toronto, ON M5S 1A1, Canada

## ABSTRACT

Current state-of-the-art methods for out-of-distribution (OOD) generalization lack the ability to effectively address datasets with heterogeneous causal subgraphs at the instance level. Existing approaches that attempt to handle such heterogeneity either rely on data augmentation, which risks altering label semantics, or impose causal assumptions whose validity in real-world datasets is uncertain. We introduce **DiSCO**, a novel *Mixture-of-Experts (MoE)* framework for **Di**versity- and **S**parsity-driven **C**ausal **O**OD graph learning, designed to model heterogeneous causal subgraphs without relying on restrictive assumptions. Our key idea is to address instance-level heterogeneity by enforcing semantic *diversity* among experts, each generating a distinct causal subgraph, while a learned gate assigns *sparse* weights that adaptively focus on the most relevant experts for each input. Our theoretical analysis shows that these two properties jointly reduce OOD error. In practice, our experts are scalable and do not require environment labels. Empirically, our framework achieves strong performance on the GOOD benchmark across both synthetic and real-world structural shifts.

## 1 INTRODUCTION

Out-of-distribution (OOD) generalization concerns learning models that remain reliable when the test distribution differs from training (Ye et al., 2021). Despite recent progress, it remains a fundamental challenge, especially for graphs (Fan et al., 2023), whose complexity admits shifts not only in node or edge attributes but also in structural properties such as size, sparsity, or motif frequency (Wu et al., 2018). Such structural shifts can mislead models, as spurious topological correlations often dominate learning (Gui et al., 2022).

A dominant paradigm in graph OOD learning is *causal subgraph* identification: each graph is assumed to contain a subgraph $G_c$ responsible for the label, while the remaining structure $G_s$ reflects spurious variation (Gui et al., 2023; Sui et al., 2025; An et al., 2024; Yao et al., 2025; Chen et al., 2022). In principle, recovering $G_c$ should yield predictions robust to distribution shifts. In practice, however, existing methods rely on restrictive causal assumptions (e.g., $G_s$ is independent of the label) that may fail in practice (Gui et al., 2022; 2023). For example, in sentiment analysis, stylistic markers such as word length can frequently track sentiment, meaning $G_s$ is not independent of the label.

A further challenge is *instance heterogeneity*: even within a single environment and label class, different samples may rely on fundamentally different causal subgraphs. For example, in molecular property prediction, multiple chemotypes can produce the same biological activity, meaning two "active" molecules may depend on entirely different causal subgraphs (Wu et al., 2018). Methods that assume a single invariant $G_c$ across environments or labels cannot capture such variability and therefore struggle under instance-level causal diversity (Sui et al., 2025). Some approaches attempt to approximate this heterogeneity via data augmentation (Wu et al., 2024; Sui et al., 2023), but perturbing graph structure cannot guarantee label correctness and may change the true causal subgraph—especially in motif-centric datasets such as GOOD-Motif, where labels correspond directly

to specific subgraphs (Gui et al., 2022; Wu et al., 2018). These limitations motivate methods that directly model causal diversity at the instance level, rather than approximating it through augmentation.

We address this gap by introducing **DiSCO**, a *Mixture-of-Experts (MoE)* framework for **Di**versity- and **S**parsity-driven **C**ausal **O**OD graph learning. Unlike prior MoE approaches that rely on predefined shift types or augmentation strategies, **DiSCO** employs experts as causal subgraph extractors, enabling them to specialize in distinct causal mechanisms. A learned gating network adaptively selects the most relevant experts for each instance. Our theoretical analysis shows that (i) *semantic diversity* is necessary for meaningful specialization, and (ii) *instance-level sparsity* naturally follows from diversity via induced loss gaps. Together, these results provide a principled justification for MoE in OOD graph learning. Importantly, our framework does not rely on environment labels or causal assumptions, making it assumption-light. Building on this foundation, our contributions are threefold:

1. **Theory:** We provide a principled justification for MoE in graph OOD learning by deriving a formal risk bound, showing that *semantic diversity* among experts and *instance-level sparsity* during gating jointly reduce OOD error.

2. **Implementation:** We design a causal subgraph-based MoE framework that instantiates these principles through a decorrelation regularizer for diversity and a learnt gating mechanism for sparsity, without requiring environment labels or strong causal assumptions.

3. **Empirics:** On the GOOD benchmark (Gui et al., 2022), **DiSCO** achieves strong performance across synthetic and real-world structural shifts. Ablations verify the necessity of diversity and sparsity.

## 2 BACKGROUND

**Graph OOD Learning.** In graph learning, each input is a graph $x$ with an associated label $y$. In an OOD setting, models are trained on data drawn from certain environments but are expected to perform well when evaluated on new environments that were not seen during training. We focus on *covariate shift*, where the relationship between $x$ and $y$ is stable but the distribution of graph inputs changes. This reflects realistic cases where the causal mechanism is preserved, yet nuisance factors such as graph size, sparsity, or motif frequency vary.

**Causal subgraph paradigm.** A common basis for graph OOD learning is the *causal subgraph paradigm*: each graph input contains a causal subgraph $G_c$ that determines the label $y$, while the complement $G_s$ captures spurious variation. Successfully identifying $G_c$ would yield shift-invariant predictions, but since $G_c$ is unobserved, existing methods rely on assumptions about the underlying structural causal model (SCM) (Gui et al., 2023; Sui et al., 2025; Chen et al., 2022; An et al., 2024; Yao et al., 2025; Wu et al., 2022; Miao et al., 2022). Examples include assuming $G_s \perp y$ (Gui et al., 2023) or that $G_c$ is invariant across environments or classes (Sui et al., 2025). Such assumptions are often unrealistic: Such assumptions are often unrealistic: they overlook *instance heterogeneity*, and they may fail when $G_s$ correlates with $y$ (e.g., molecular scaffolds shared by active compounds (Zhang et al., 2024)).

**Instance heterogeneity.** Instance heterogeneity refers to the setting where the causal subgraph varies not only across environments but also across individual samples within the same dataset and label class. Formally, if each graph $G$ contains a (possibly latent) causal subgraph $G_c$ sufficient to predict $y$, then the collection of such $G_c$'s need not be identical across the dataset; this variability constitutes causal diversity. Such diversity arises naturally in real-world domains and is also present in the GOOD benchmark (Gui et al., 2022). In GOOD-HIV, multiple chemotypes can yield the same biological activity, and in GOOD-SST2, sentiment can be expressed through structurally distinct syntactic patterns. As a result, two instances from the same environment with the same label may rely on fundamentally different causal subgraphs, and methods that assume a single invariant $G_c$ across environments or classes cannot account for such causal diversity. While some works attempt to approximate heterogeneity via data augmentation (Lu et al., 2024; Yao et al., 2024; Wu et al., 2024; Chen et al., 2023; Sui et al., 2023; Miao et al., 2022), perturbations do not guarantee label correctness and may alter the true causal subgraph—especially in motif-centric datasets such as GOOD-Motif (Gui et al., 2022). These limitations motivate the need for frameworks that directly

model causal diversity at the instance level. We propose an alternative perspective: model heterogeneity directly through a *MoE framework* by allowing multiple experts to extract diverse causal subgraphs and using instance-specific gating to focus on the most relevant ones.

**Causal Assumptions.** Many OOD graph methods rely on assumptions about the underlying SCM, particularly regarding how the causal subgraph $G_c$, spurious subgraph $G_s$, and label $y$ interact. A common assumption is that $G_s$ is conditionally independent of $y$ given $G_c$ (Gui et al., 2023; Sui et al., 2025; Chen et al., 2022; An et al., 2024; Yao et al., 2025; Wu et al., 2022; Miao et al., 2022). However, this assumption does not hold under alternative SCMs such as FIIF (fully informative invariant features), where $G_s$ is entirely determined by $G_c$, or PIIF (partially informative invariant features), where $G_s$ is partly driven by $G_c$ but still correlated with $y$. In practice, such independence conditions are rarely guaranteed. In molecular property prediction, for instance, structural scaffolds often correlate with biological activity, violating $G_s \perp y$. In social or textual graphs, stylistic markers frequently track sentiment, creating direct $G_s \rightarrow y$ dependencies. Even in controlled benchmarks such as the GOOD benchmark (Gui et al., 2022), these SCM-level assumptions are *not* guaranteed. Consequently, relying on a single causal assumption can lead to brittle behavior. In contrast, our method imposes no SCM-level assumptions: we avoid invariance-based losses and instead allow instance-specific expert selection, yielding robustness to diverse causal mechanisms.

**MoE for GNNs.** MoE architectures (Jacobs et al., 1991; Shazeer et al., 2017) consist of multiple experts combined through a gating function, and in graph domains they have primarily been applied to improve scalability and efficiency (Wang et al., 2023a; Chen et al., 2025; Hu et al., 2022). To the best of our knowledge, GraphMETRO is the only OOD-oriented MoE approach to date, assigning each expert to a predefined shift type and training them as augmentation strategies invariant to those shifts (Wu et al., 2024). This approach naturally inherits the label validity risks mentioned above. In contrast, we employ MoE for *causal subgraph identification*, encouraging experts to capture diverse causal hypotheses and directly addressing instance heterogeneity. Additional related work is discussed in Appendix B.

## 3 METHOD

In this section, we derive an explicit risk bound that decomposes OOD error into *coverage* and *selection* terms, and we show how *diversity* and *sparsity* jointly serve to reduce it. All proofs are deferred to Appendix A.

### 3.1 PRELIMINARIES AND NOTATION

We consider supervised learning on graphs, where each instance is $x = (V, E, X)$ with node set $V$, edge set $E$, and node features $X$, together with an associated label $y \in \mathcal{Y}$. Let $\mathcal{X}$ denote the input space of graphs and $\mathcal{D}$ a distribution over $(x, y) \in \mathcal{X} \times \mathcal{Y}$. We write $\mathcal{D}_{\mathcal{X}}$ for the marginal on $\mathcal{X}$. A predictor $h$ maps $x$ to logits $z_h(x) \in \mathbb{R}^C$, where $C = |\mathcal{Y}|$ is the number of classes and $c \in \{1, \ldots, C\}$ indexes a class. Predictions are evaluated with the cross-entropy loss $\ell_{\mathrm{CE}}(z, y)$ (Goodfellow et al., 2016), and the risk of $h$ under distribution $\mathcal{D}$ is

$$R_{\mathcal{D}}(h) = \mathbb{E}_{(x,y)\sim\mathcal{D}}[\ell_{\mathrm{CE}}(z_h(x), y)].$$

We define OOD generalization with respect to a family of environments $\mathcal{M}$, where each $m \in \mathcal{M}$ corresponds to a distribution $\mathcal{D}_m$ over $\mathcal{X} \times \mathcal{Y}$. Training occurs on a subset $\mathcal{M}_{\mathrm{train}}$, while evaluation is on unseen test environments $\mathcal{M}_{\mathrm{test}}$ with $\mathcal{M}_{\mathrm{test}} \setminus \mathcal{M}_{\mathrm{train}} \neq \emptyset$. We denote a particular unseen test environment by $m'$, with associated distribution $\mathcal{D}_{m'}$. The objective is to control the worst-case risk

$$R_{\mathrm{OOD}}(h) = \sup_{m \in \mathcal{M}_{\mathrm{test}}} R_{\mathcal{D}_m}(h).$$

We focus on the case of *covariate shift*, where the marginal distribution over $\mathcal{X}$ varies across environments while the conditional distribution $P(y \mid x)$ remains stable.

**Mixture-of-Experts.** Let $\{h_i\}_{i=1}^K$ denote $K$ experts, and let $\pi(x) \in \Delta^K$ be a gating distribution (sometimes we omit the dependence on $x$ and write simply $\pi$). Each expert $h_i$ outputs logits $z_i(x) \in \mathbb{R}^C$. The MoE aggregates predictions via

$$z_{\mathrm{mix}}(x) = \sum_{i=1}^K \pi_i(x)\, z_i(x), \qquad \hat{y}(x) = \arg\max_{c \in \{1, \ldots, C\}} z_{\mathrm{mix}}(x)[c].$$

For expert $h_i$, the cross-entropy loss is $\ell_i(x, y) = \ell_{\mathrm{CE}}(z_i(x), y)$, and the MoE risk under distribution $\mathcal{D}$ is

$$R_{\mathcal{D}}(\mathrm{MoE}) = \mathbb{E}_{(x,y)\sim\mathcal{D}}[\ell_{\mathrm{CE}}(z_{\mathrm{mix}}(x), y)].$$

We also define the auxiliary mixture-of-losses quantity

$$\bar{\ell}(x, y) := \sum_{i=1}^{K} \pi_i(x)\, \ell_i(x, y),$$

which by Jensen's inequality (Jensen, 1906) upper-bounds the mixture-of-predictions loss.

## 3.2 Diversity between experts (coverage)

For a MoE model to be effective, experts must represent distinct hypotheses; otherwise the gate has no meaningful choice and the model collapses. We therefore encourage *semantic diversity*: experts should emphasize different subgraphs in the input, providing *coverage* over latent causal mechanisms. As we will show, such diversity not only prevents model collapse but also induces non-trivial specialization, which in turn forces sparse gating.

**Definition 3.1** (Semantic diversity). *Let expert $i$ produce a mask probability vector $\mathbf{v}_i^{(x)} \in [0, 1]^{|I_x|}$ on input graph $x$ (nodes/edges indexed by $I_x$). Standardize*

$$\tilde{\mathbf{v}}_i^{(x)} = \frac{\mathbf{v}_i^{(x)} - \mu_i^{(x)}\mathbf{1}}{\sigma_i^{(x)} + \varepsilon}, \quad \mu_i^{(x)} = \tfrac{1}{|I_x|}\sum_{u\in I_x} \mathbf{v}_i^{(x)}(u), \quad (\sigma_i^{(x)})^2 = \tfrac{1}{|I_x|}\sum_{u\in I_x}\left(\mathbf{v}_i^{(x)}(u) - \mu_i^{(x)}\right)^2,$$

*with a small $\varepsilon > 0$ to avoid division by zero. Let $\rho_{ij}^{(x)} = \frac{1}{|I_x|}\langle \tilde{\mathbf{v}}_i^{(x)}, \tilde{\mathbf{v}}_j^{(x)}\rangle$. We say the experts are semantically diverse if*

$$\frac{1}{K(K-1)} \sum_{i\neq j} \mathbb{E}_{x\sim\mathcal{D}_{\mathcal{X}}}\big[|\rho_{ij}^{(x)}|\big] \leq \tau_{corr} \quad \text{for some } \tau_{corr} > 0.$$

By enforcing low correlation between experts' masks, semantic diversity ensures experts attend to different parts of the input graph. Under standard GNN encoders whose logits are Lipschitz in masked embeddings (Scarselli et al., 2008; Joshi et al., 2023), low mask correlation encourages encoding of distinct structural signals, discouraging collapse onto the same subgraph. This guarantees *coverage* of multiple causal hypotheses, giving the MoE leverage to identify causal subgraphs under heterogeneity.

## 3.3 Instance-level sparsity (selection)

If one expert predicts the causal mechanism for an input while others do not, the mixture is reliable only if the gate concentrates sufficient mass on that expert: the *selection* effect. This ensures that once distinct causal mechanisms exist among the experts (via semantic diversity), the gate can isolate the correct one. To formalize this, we introduce a metric, the *loss gap*, which measures the discrepancy between the best and next-best expert on an instance.

**Definition 3.2** (Loss gap). *For $(x, y)$, let $i^\star(x, y) = \arg\min_{i\in[K]} \ell_i(x, y)$ denote the best expert, where $[K] = \{1, \ldots, K\}$. The loss gap is*

$$\Delta(x, y) = \min_{j\neq i^\star(x,y)} \big(\ell_j(x, y) - \ell_{i^\star}(x, y)\big) \geq 0.$$

We now show that loss gaps quantitatively constrain the gate's allocation.

**Proposition 3.3** (Loss gap implies sparsity). *Let $\{\ell_i(x, y)\}_{i=1}^{K}$ be the per-expert losses and $i^\star(x, y) \in \arg\min_{i\in[K]} \ell_i(x, y)$ be any minimizer. Define the mixture-of-losses $\bar{\ell}(x, y) = \sum_{i=1}^{K} \pi_i(x)\, \ell_i(x, y)$ and the loss gap*

$$\Delta(x, y) := \begin{cases} \min_{k\neq i^\star(x,y)} \big(\ell_k(x, y) - \ell_{i^\star}(x, y)\big), & K \geq 2, \\ 0, & K = 1. \end{cases}$$

*Then*

$$\bar{\ell}(x,y) \geq \ell_{i^\star}(x,y) + \left(1 - \pi_{i^\star}(x)\right) \Delta(x,y).$$

*Equivalently, for any $\Delta(x,y) > 0$,*

$$\pi_{i^\star}(x) \geq 1 - \frac{\bar{\ell}(x,y) - \ell_{i^\star}(x,y)}{\Delta(x,y)}.$$

This bound shows that to keep the mixture loss close to the best expert, the gate must assign sufficient weight to that expert. Furthermore, *larger loss gaps make this requirement stronger*. Such loss gaps are expected to arise as a consequence of diversity, which forces experts to attend to decorrelated subgraphs. We formalize this in the following assumption:

**Assumption 3.4** (Diversity induces loss gaps)**.** *Let $\Delta(x,y)$ be as in Definition 3.2. There exist $\gamma > 0$, $\rho \in [0,1)$, and a measurable set $\mathcal{S} \subseteq \mathcal{X} \times \mathcal{Y}$ with $\mathcal{D}(\mathcal{S}) \geq 1 - \rho$ such that*

$$\Delta(x,y) \geq \gamma \quad \forall (x,y) \in \mathcal{S},$$

*and on $\mathcal{S}$ the minimizer $i^\star(x,y)$ coincides with an expert aligned with the environment's causal mechanism. Semantic diversity (Definition 3.1) promotes this condition by forcing experts to attend to decorrelated subgraphs.*

This assumption formalizes the idea that, under semantic diversity, one causal expert should outperform others by a margin on most inputs. Intuitively, if one expert captures the causal subgraph, then by semantic diversity, the others focus on decorrelated (likely spurious) subgraphs, yielding non-trivial loss gaps. This is validated empirically in Section 4. Hence, with sufficient diversity, loss gaps should arise on a large fraction of inputs, and the MoE can match the best expert only by *selecting* it via sparse gating: *diversity necessitates sparsity*.

### 3.4 AN OOD RISK DECOMPOSITION: COVERAGE AND SELECTION

We now show how these two properties jointly control OOD risk by decomposing risk into a coverage term (controlled by diversity) and a selection term (controlled by sparsity). First, we make the following assumption:

**Assumption 3.5** (Mechanism coverage)**.** *Let $\{\mathcal{D}_m : m \in \mathcal{M}\}$ denote environments and $h_{m'}^\star \in \arg\min_h R_{\mathcal{D}_{m'}}(h)$ the oracle predictor for environment $\mathcal{D}_{m'}$. For any OOD environment $\mathcal{D}' = \mathcal{D}_{m'}$, there exists an expert $i^\star(m') \in [K]$ (depending only on the environment) such that*

$$R_{\mathcal{D}_{m'}}(h_{i^\star(m')}) \leq R_{\mathcal{D}_{m'}}(h_{m'}^\star) + \varepsilon_{\text{cov}}(m').$$

That is, for every unseen environment, at least one expert *covers* it by achieving risk within $\varepsilon_{\text{cov}}(m')$ of the oracle predictor. To minimize OOD risk, the MoE must then *select* this expert by allocating enough probability to it, yielding:

**Theorem 3.6** (OOD risk: coverage + selection)**.** *Fix an OOD environment $\mathcal{D}' = \mathcal{D}_{m'}$. For the environment-aligned expert $i^\star(m')$, define*

$$\Gamma_{m'}(x,y) := \max_{j \neq i^\star(m')} \left(\ell_j(x,y) - \ell_{i^\star(m')}(x,y)\right).$$

*Then, under Assumptions 3.5 and 3.4,*

$$R_{\mathcal{D}'}(\text{MoE}) \leq \underbrace{R_{\mathcal{D}'}(h_{m'}^\star)}_{\text{oracle risk}} + \underbrace{\varepsilon_{\text{cov}}(m')}_{\text{coverage via diversity}} + \underbrace{\mathbb{E}_{(x,y) \sim \mathcal{D}'}\left[(1 - \pi_{i^\star(m')}(x)) \Gamma_{m'}(x,y)\right]}_{\text{selection penalty via sparsity}}.$$

This decomposition expresses OOD risk as three parts. The first is the *oracle risk*, the irreducible error in the target environment. The second is the *coverage term*, kept small by semantic diversity: only when experts are sufficiently diverse can the pool cover unseen test environments by ensuring at least one aligns with the causal mechanism. The third is the *selection penalty*, kept small by sparsity: loss gaps make the aligned expert identifiable, but only a sparse gate can reliably concentrate on it. Together, these conditions show that to jointly reduce OOD error, *diversity is needed to ensure a good expert exists, and sparsity is needed to ensure that it is selected*.

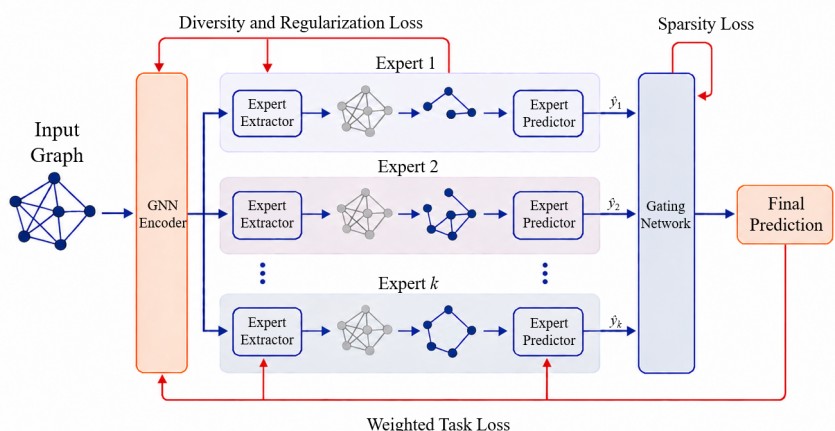

Figure 1: An illustration of the proposed MoE architecture. Each expert extracts a distinct candidate causal subgraph and produces predictions, which are combined by the gating network into the final output.

## 3.5 IMPLEMENTATION

A detailed illustration of our implementation is shown in Figure 1. Given an input graph $x = (V, E, X)$, we first compute node embeddings with a shared GNN encoder. For each expert $i \in [K]$, a small expert-specific MLP takes the concatenated embeddings of edge endpoints and outputs a mask logit $\ell_e^{(i)}$ for every edge $e \in E$, which is later transformed into a binary selection through the Gumbel–sigmoid straight-through estimator for differentiability (Maddison et al., 2017; Jang et al., 2017). The resulting masked graph $x^{(i)}$, representing the extracted causal subgraph by expert $i$, is then passed through an expert-specific GNN and classifier head to produce logits $z_i(x^{(i)})$. Predictions are combined by a lightweight MLP gate that consumes expert-derived statistics (e.g., confidence, entropy) and outputs a weight vector $\pi(x) \in \Delta^K$ over experts. The final prediction is obtained as the weighted average of expert logits using $\pi(x)$. The entire model is trained end-to-end with a combination of task, regularization, diversity, and gating losses.

**Task loss.** For each expert $i$, the overall task loss is

$$\ell_{\text{CE}}(x, y) = \sum_i \pi_i(x) \, \ell_i(x, y),$$

where $\ell_i(x, y)$ is the per-sample cross-entropy for expert $i$. By weighting the loss with the gate probabilities, experts that contribute more to the final prediction receive stronger gradients, while those assigned little weight are suppressed, promoting specialization.

**Regularization Loss.** To control the size of extracted subgraphs, we regularize the average fraction of edges retained by each expert. For expert $i$ on input $x$, let $\rho_i^{(x)}$ denote its observed keep-rate. We then penalize deviations from a target $\rho \in [0, 1]$ using

$$\ell_{\text{reg}}^{(i)} = (\rho_i^{(x)} - \rho)^2,$$

which discourages degenerate solutions where experts keep either too few or too many edges.

**Diversity loss.** To prevent experts from collapsing onto identical subgraphs, we compute

$$\ell_{\text{div}} = \frac{1}{K(K-1)} \sum_{i \neq j} \max\{0, |\rho_{ij}^{(x)}| - \tau_{\text{corr}}\},$$

where $\rho_{ij}^{(x)}$ denotes the correlation between the masks of experts $i$ and $j$ on input $x$ and the model is penalized when this correlation exceeds threshold $\tau_{\text{corr}}$. This directly follows from the semantic diversity condition (Definition 3.1) and drives experts to specialize on distinct subgraphs, in turn ensuring coverage.

**Gating loss.** We train the gate with a teacher–student objective. The teacher distribution $q$ is defined by normalizing the negative per-expert cross-entropy losses, giving higher weight to lower-loss experts, while the student distribution is the gate output $p = \pi(x)$. This alignment teaches the gate which experts are competent on which inputs. Since the task loss is gate-weighted, non-selected experts are not heavily penalized, reinforcing specialization rather than forcing uniformity among experts. To further shape the gate, we add sparsity and balance regularizers, yielding the objective

$$\ell_{\text{gate}} = \text{KL}(p\|q) + \lambda_{\text{sparse}}\,\ell_{\text{sparse}}(p) + \lambda_{\text{bal}}\,\ell_{\text{bal}}(p).$$

Here, $\ell_{\text{sparse}}$ penalizes high-entropy distributions to enforce instance-level *sparsity*, while $\ell_{\text{bal}}$ encourages even usage across the batch to avoid expert starvation, supporting *coverage*. These terms are complementary: sparsity sharpens routing for each input, while balance spreads usage globally. For stability, training begins with a short warm-up phase of uniform routing so all experts receive sufficient training signals before specialization. The gate can be fine-tuned after experts have been trained to better align with the learned specializations.

**Total loss.** The final training objective is

$$\mathcal{L} = \ell_{\text{CE}} + \lambda_{\text{reg}}\ell_{\text{reg}} + \lambda_{\text{div}}\ell_{\text{div}} + \lambda_{\text{gate}}\ell_{\text{gate}}.$$

This loss function implements the components of the theoretical framework while incorporating additional regularizers that prevent degenerate solutions and expert starvation.

**Theoretical computational complexity.** Our MoE architecture is designed to introduce only modest computational overhead. As illustrated in Figure 1, all experts share a single subgraph extractor, and only lightweight per-expert output heads (MLPs) are replicated. Each expert consists of one GNN encoder followed by a small classifier (MLP), in contrast to existing OOD graph methods that require multiple full GNNs for classification (Gui et al., 2023; An et al., 2024).

Theoretically, if a GNN forward pass costs $\mathcal{O}(f(G))$ and each MLP forward pass costs $\mathcal{O}(g(G))$, then the total complexity of our MoE is:

$$\mathcal{O}\big(f(G) + K\,g(G) + K\,f(G) + K\,g(G) + g(G)\big),$$

corresponding respectively to the shared extractor, per-expert heads, per-expert GNNs, per-expert classifiers, and the gating network. Since GNN computation dominates MLP computation, $f(G) \gg g(G)$ (Goodfellow et al., 2016; **?**), this simplifies to:

$$\mathcal{O}(K\,f(G)),$$

indicating that the overhead grows linearly in the number of experts $K$ and is dominated by the shared GNN encoder. In Appendix G, we provide empirical analyses on the training and inference time of our method.

**Assumption-light design without auxiliary invariance losses.** Many prior OOD graph methods learn causal subgraphs through auxiliary objectives (e.g., adversarial discriminators in LECI (Gui et al., 2023), structural alignment in UIL (Sui et al., 2025)). While effective in single-model settings, these approaches are *computationally expensive* when replicated across experts and may be *unstable* (e.g., requiring finely tuned adversarial schedules that vary across experts). More critically, they rely on *restrictive causal assumptions* (e.g., $G_s \perp Y$, invariant $G_c$) that rarely hold in heterogeneous real-world data (Zhang et al., 2024). By contrast, our design avoids such auxiliary losses entirely. As a result, it scales gracefully with the number of experts and requires neither environment labels nor restrictive causal assumptions. This key design choice makes our method both highly scalable and assumption-light. We provide further discussion in Appendix D.

## 4 EXPERIMENTS

We now evaluate our method empirically, guided by five research questions: **RQ1**: Does the method achieve strong performance on both synthetic and real-world datasets with structural shifts? **RQ2**: Does the method achieve strong performance across different causal assumptions? **RQ3**: Are the key components of the framework necessary to obtain the reported improvements? **RQ4**: Does enforcing semantic diversity induce larger loss gaps and promote expert specialization? **RQ5**: How sensitive is performance to hyperparameters?

Table 1: **Results on graph classification datasets with structural shifts from the GOOD benchmark.** Values are classification accuracy (ROC-AUC for HIV) on OOD test sets, averaged over 5 runs with standard deviation in parentheses. Table sections correspond to domain generalization, data augmentation, and causal subgraph methods. We also report average performance and rank across datasets. Best results are in **bold**. † indicates methods requiring environment labels.

| Method | HIV ↑ | | Twitter ↑ | Motif ↑ | | SST2 ↑ | Avrg ↑ | Avrg Rank ↓ |
| | Scaffold | Size | Length | Basis | Size | Length | | |
|---|---|---|---|---|---|---|---|---|
| ERM | 69.89 (1.95) | 58.12 (2.5) | 58.56 (1.0) | 64.32 (10.4) | 54.29 (5.4) | 80.23 (0.8) | 64.24 | 10.83 |
| Coral | 72.33 (2.1) | 60.33 (3.5) | 57.33 (0.9) | 65.39 (9.6) | 52.39 (2.9) | 79.23 (1.8) | 64.50 | 10.33 |
| IRM† | 71.11 (2.7) | 60.67 (1.4) | 57.79 (1.9) | 62.64 (10.9) | 54.14 (5.2) | 80.37 (1.7) | 64.45 | 10.33 |
| VREx† | 70.94 (3.1) | 61.10 (3.0) | 56.55 (0.7) | 65.13 (5.0) | 56.97 (6.3) | 79.85 (1.6) | 65.09 | 10.83 |
| GDRO† | 67.13 (2.3) | 56.91 (3.0) | 56.73 (0.9) | 62.63 (8.9) | 52.01 (3.6) | 81.33 (0.9) | 62.79 | 12.50 |
| DANN† | 67.69 (2.9) | 62.05 (2.1) | 56.09 (1.7) | 52.65 (6.8) | 49.33 (5.4) | 80.59 (0.9) | 61.40 | 13.00 |
| GM | 66.06 (4.0) | 66.24 (2.9) | 56.97 (3.1) | 67.33 (5.9) | 61.43 (6.5) | 81.96 (0.6) | 66.67 | 7.83 |
| AIA | 71.23 (1.4) | 62.33 (4.6) | 57.13 (1.8) | 74.18 (5.9) | 56.07 (5.3) | 80.91 (1.0) | 66.98 | 7.67 |
| GALA | **74.51 (1.8)** | 64.89 (1.7) | 60.79 (0.7) | 79.11 (3.2) | 72.13 (1.4) | 82.42 (0.7) | 72.31 | 2.67 |
| LIRS | 70.70 (2.3) | 64.46 (2.9) | 58.76 (1.4) | 74.16 (3.0) | 72.61 (6.9) | 81.20 (0.7) | 70.32 | 5.83 |
| GSAT | 70.76 (1.5) | 61.76 (2.1) | 57.13 (0.8) | 62.27 (0.8) | 54.12 (5.2) | 80.62 (0.5) | 64.44 | 10.67 |
| CIGA | 71.33 (1.1) | 63.09 (1.6) | 58.01 (2.2) | 38.01 (1.4) | 55.69 (6.7) | 80.56 (1.7) | 61.12 | 9.17 |
| DIR | 68.06 (5.5) | 61.22 (0.8) | 57.19 (0.9) | 36.10 (2.5) | 43.98 (3.1) | 81.13 (0.7) | 57.95 | 12.17 |
| LECI† | 74.28 (1.7) | 65.76 (1.4) | 59.90 (0.2) | 85.74 (3.0) | 71.92 (1.4) | 83.27 (0.3) | 73.48 | 2.67 |
| UIL† | 62.51 (1.7) | 64.79 (0.8) | 59.66 (0.9) | 61.77 (4.8) | 68.47 (3.1) | 82.03 (0.4) | 66.54 | 7.83 |
| **DiSCO** | 71.55 (1.4) | **66.98 (1.0)** | **61.13 (1.1)** | **92.80 (1.4)** | **75.52 (2.9)** | **83.73 (1.4)** | **75.29** | **1.50** |

**Datasets.** We evaluate on the GOOD benchmark (Gui et al., 2022), which provides training, OOD validation, and OOD test splits. Our study covers six datasets with structural shifts: HIV-Scaffold/Size (molecular), Motif-Basis/Size (synthetic motifs), Twitter-Length (social), and SST2-Length (sentiment). These span both synthetic and real-world domains under diverse structural covariate shifts. We also evaluate on CFP-Motif (Gui et al., 2023), which provides datasets with three different causal assumptions: covariate, FIIF, and PIIF. Unless otherwise specified, results are averaged over five seeds.

**Hyperparameters.** All models adopt the Graph Isomorphism Network (GIN) (Xu et al., 2019), the default GOOD backbone, with standard hyperparameters. Unless otherwise noted, we use eight experts within our MoE model. We tune mask keep-rate prior $\rho$, batch size, and learning rate over 10 trials on the OOD validation set.

**Baselines.** We compare against three groups of methods. (1) General domain generalization algorithms: ERM, IRM (Arjovsky et al., 2019), Coral (Sun & Saenko, 2016), V-REx (Krueger et al., 2021), GroupDRO (GDRO for brevity) (Sagawa et al., 2020), and DANN (Ganin et al., 2016). (2) Data augmentation methods that address instance heterogeneity: GraphMETRO (GM for brevity) (Wu et al., 2024), AIA (Chen et al., 2023), and GALA (Sui et al., 2023). (3) Causal subgraph algorithms: LIRS (Yao et al., 2025), GSAT (Miao et al., 2022), CIGA (Chen et al., 2022), DIR (Wu et al., 2022), LECI (Gui et al., 2023), and UIL (Sui et al., 2025). All results are reproduced using official repositories and hyperparameter settings.

## 4.1 RESULTS AND DISCUSSION

**RQ1: Does the method achieve strong performance on both synthetic and real-world datasets with structural shifts?** On the synthetic datasets (HIV-Scaffold/Size and Motif-Basis/Size), our method achieves the best performance except on HIV–Scaffold. Notably, on Motif–Basis, a common sanity test for causal subgraph methods since labels are determined by the presence of specific motifs (Gui et al., 2022; 2023), our method achieves 92.8% accuracy which is comparable to oracle-level performance (Gui et al., 2023), and an 8.2% relative improvement over the next best method. On HIV–Scaffold, the severe class imbalance (over 95% majority class) makes the task especially challenging, and our method attains 71.55 ROC-AUC, ranking fourth overall. On the real-world datasets (Twitter and SST2), our method outperforms all baselines, demonstrating robustness under

Table 2: Results on different causal assumption datasets from CFP-Motif. Values are are classification accuracy with standard deviation in parentheses. Baseline results are from Gui et al. (2023). Best results are in **bold**.

| | Covariate ↓ | FIIF ↓ | PIIF ↓ |
|---|---|---|---|
| ERM | 57.56 (9.59) | 37.22 (3.70) | 62.45 (9.21) |
| IRM | 58.11 (5.14) | 44.33 (1.52) | 68.34 (10.40) |
| VREx | 48.78 (7.81) | 34.78 (1.34) | 63.33 (6.55) |
| Coral | 57.11 (8.35) | 42.68 (7.09) | 60.33 (8.85) |
| DANN | 49.45 (8.05) | 43.22 (6.64) | 62.56 (10.39) |
| DIR | 44.67 (0.00) | 42.00 (6.77) | 47.22 (8.79) |
| GSAT | 68.22 (7.23) | 51.56 (6.59) | 61.22 (8.80) |
| CIGA | 56.78 (2.99) | 39.11 (7.70) | 45.67 (7.52) |
| LECI | 83.20 (5.89) | 77.73 (3.85) | 69.40 (7.54) |
| **DiSCO** | **90.83 (1.73)** | **84.17 (3.88)** | **77.19 (6.04)** |

Table 3: Performance of the MoE framework on the GOOD benchmark under ablations of each loss term. Standard deviations are reported in parentheses.

| Dropping | HIV ↑ | | Twitter ↑ | Motif ↑ | | SST2 ↑ |
|---|---|---|---|---|---|---|
| | Scaffold | Size | Length | Basis | Size | Length |
| $\ell_{\text{div}}$ | 65.95 (2.5) | 65.23 (0.8) | 60.10 (0.9) | 91.13 (1.1) | 70.07 (2.8) | 82.20 (1.3) |
| $\ell_{\text{gate}}$ | 68.56 (0.9) | 61.91 (1.2) | 59.92 (0.5) | 89.60 (2.7) | 74.36 (2.0) | 81.97 (1.2) |
| $\ell_{\text{reg}}$ | 68.55 (3.9) | 64.79 (2.2) | 60.60 (1.9) | 67.48 (7.0) | 73.75 (2.5) | 83.46 (1.5) |
| Standard (full loss) | **71.55 (1.4)** | **66.98 (1.0)** | **61.13 (1.1)** | **92.80 (1.4)** | **75.52 (2.9)** | **83.73 (1.4)** |

noisy, real-world distribution shifts. Across all datasets, our method achieves the *highest average score (75.29) and lowest average rank (1.5)*. The closest competitors are LECI (73.48 average, 2.67 rank) and GALA (72.31 average, 2.67 rank), confirming substantial improvements over existing baselines.

**RQ2: Does the method achieve strong performance across different causal assumptions?** On the CFP-Motif dataset (Table 2), our method achieves the best performance under all three causal assumptions, with LECI as the strongest baseline. We attribute this improvement to avoiding auxiliary invariance losses that implicitly enforce a fixed causal assumption. By not committing to a particular SCM, our method adapts more effectively to the different causal assumptions present in CFP-Motif.

**RQ3: Are the key components of the framework necessary for the reported improvements?** Table 3 presents an ablation study on the GOOD benchmark, evaluating the contribution of each major component: *semantic diversity*, *instance-level sparsity*, and *regularization*. We systematically remove the corresponding loss terms and observe substantial performance degradations across most datasets, with particularly large drops on Motif-Basis, HIV-Scaffold, and HIV-Size. These results indicate that all three components are essential for achieving the full performance gains of our framework.

**RQ4: Does enforcing semantic diversity induce larger loss gaps and promote expert specialization?** To evaluate Assumption 3.4 (semantic diversity induces loss gaps), we compute the total loss gap within each batch and then average it over the entire test set. We then compare models trained with and without the diversity objective. Table 4 shows that semantic diversity consistently increases the average per-batch loss gap across datasets. On Twitter, the gap rises from 0.13 to 0.19 (a 46% increase); on SST2, from 0.07 to 0.22 (over 200% increase); and on Motif-Basis, from 0.076 to 0.12 (a 58% increase). These results empirically validate our assumption that semantic diversity among experts induces larger loss gaps. We provide additional evidence in Appendix F.1, which demonstrates diverse expert specialization across causal subgraphs, and in Appendix F.3, which presents visualizations of the causal subgraphs extracted by each expert.

**RQ5: How sensitive is performance to hyperparameters?** We first evaluate the effect of the edge keep-rate prior $\rho$ on Twitter, SST2, and Motif-Basis by perturbing it around the tuned value (shown

Table 4: Total loss gaps per-batch under ablations of diversity on Twitter, SST2, and Motif-Basis. Standard deviations are reported in parentheses.

| Dataset | Diversity | Loss Gap |
|---|---|---|
| Twitter | w/ | 0.19 (0.011) |
| | w/o | 0.13 (0.004) |
| SST2 | w/ | 0.22 (0.12) |
| | w/o | 0.07 (0.19) |
| Motif-Basis | w/ | 0.12 (0.035) |
| | w/o | 0.076 (0.024) |

Table 5: Sensitivity analysis of the edge keep-rate prior $\rho$ on Twitter, SST2, and Motif-Basis. The value in brackets indicates the tuned $\rho$ selected via validation for each dataset, with performance shown at this setting and at perturbed values.

| $\rho$ | Twitter ↑ (0.55) | SST2 ↑ (0.2) | Motif-Basis ↑ (0.55) |
|---|---|---|---|
| -0.2 | 58.75 | – | 90.13 |
| -0.1 | 60.33 | 82.06 | 89.76 |
| Tuned | 61.13 | 83.73 | 92.80 |
| +0.1 | 59.51 | 83.15 | 92.03 |
| +0.2 | 60.64 | 83.46 | 90.54 |

Table 6: Performance of the MoE framework with 1, 4, and 8 experts. Standard deviations are reported in parentheses.

| Experts | HIV ↑ | | Twitter ↑ | Motif ↑ | | SST2 ↑ | Avrg ↑ |
|---|---|---|---|---|---|---|---|
| | Scaffold | Size | Length | Basis | Size | Length | |
| 1 | 67.13 (2.2) | 63.48 (0.9) | 58.68 (1.7) | 89.3 (2.1) | 65.31 (1.2) | 81.11 (1.1) | 70.84 |
| 4 | 70.65 (1.3) | 65.35 (1.3) | 60.48 (0.7) | 91.79 (1.5) | 76.4 (2.1) | 83.41 (1.2) | 74.68 |
| 8 | 71.55 (1.4) | 66.98 (1.0) | 61.13 (1.1) | 92.8 (1.4) | 75.52 (2.9) | 83.73 (1.4) | 75.29 |

in brackets). Table 5 shows that accuracy varies by at most 1–2% under shifts of ±0.1 or ±0.2, confirming robustness to moderate deviations. For example, Twitter peaks at $\rho = 0.55$ (61.13%), SST2 at $\rho = 0.20$ (83.73%), and Motif-Basis at $\rho = 0.55$ (92.80%), with nearby settings yielding comparable results. These results demonstrate that our framework is not overly sensitive to the edge keep-rate prior $\rho$. Furthermore, our hyperparameters are tuned over a modest budget of only 10 trials. We then analyze the impact of the number of experts in Table 6, which compares 1, 4, and 8 experts. Moving from 1 to 4 experts yields substantial gains across all datasets, demonstrating the importance of expert diversity and sparse gating. Increasing further to 8 experts provides smaller but generally positive improvements (except on Motif-Size). Importantly, with only 4 experts, the MoE framework achieves an average accuracy of 74.68%, outperforming all baselines in Table 1, suggesting that our method is not highly sensitive to the exact number of experts once diversity is present.

## 5 CONCLUSION

In this work, we introduce a causal subgraph–based MoE framework that explicitly addresses instance-level heterogeneity, enabling different experts to capture distinct causal explanations within the same class. Our framework demonstrates that *diversity* among experts provides coverage of heterogeneous causal mechanisms, while *sparsity* in the gating step enables effective selection, together reducing OOD error. We operationalize these principles in a scalable, assumption-light architecture that requires neither environment labels nor restrictive causal assumptions. Empirically, the method achieves strong performance on the GOOD benchmark across both synthetic and real-world shifts, with ablations and visualizations confirming that experts specialize in distinct causal mechanisms. Looking ahead, broadening causal perspectives on OOD graph learning, through richer causal mechanisms, more flexible expert designs, and closer theory–practice integration, remains an important direction for building robust and generalizable graph learning systems. We hope this work establishes MoE as a strong foundation for causal-based OOD graph learning.

### REPRODUCIBILITY STATEMENT

All mathematical proofs are provided in Appendix A. Implementation details are provided in Appendix C. The source code and reproduction instructions are available here.

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

# A   APPENDIX: PROOFS

This appendix collects the proofs of all formal results stated in the main body, together with some natural extensions. Our goal is not to introduce new theory but to provide full technical details and clarify intermediate steps that were omitted for brevity in the main text.

## A.1   SPARSITY REQUIREMENT IN AN EXTREME CASE

We begin with a simple but instructive lemma that is not included in the main body. It captures an extreme case of expert selection, where exactly one expert achieves a positive margin and all others are non-positive (that is, only one expert is correct). While not needed for the main theoretical results, we include it here because it highlights in the clearest possible terms why sparse gating is necessary: in such scenarios, correct predictions are only guaranteed if the gating mechanism concentrates sufficient mass on the correct expert.

**Lemma A.1** (Sparsity requirement in an extreme case). *Let per-expert margins be*

$$m_i(x) = z_i(x)[y] - \max_{c \neq y} z_i(x)[c],$$

*and let the mixture logits be*

$$z_{\mathrm{mix}}(x) = \sum_{i=1}^{K} \pi_i(x)\, z_i(x), \qquad \pi(x) \in \Delta^K.$$

*If there exists an expert $i^\star$ with $m_{i^\star}(x) > 0$ and $m_j(x) \leq 0$ for all $j \neq i^\star$, then the mixture margin*

$$m_{\mathrm{mix}}(x) := z_{\mathrm{mix}}(x)[y] - \max_{c \neq y} z_{\mathrm{mix}}(x)[c]$$

*is strictly positive whenever*

$$\pi_{i^\star}(x) > \alpha, \qquad \alpha = \frac{-\min_{j \neq i^\star} m_j(x)}{m_{i^\star}(x) - \min_{j \neq i^\star} m_j(x)} \in [0, 1).$$

*Proof.* We have

$$m_{\mathrm{mix}}(x) = \sum_i \pi_i(x)\, z_i(x)[y] - \max_{c \neq y} \sum_i \pi_i(x)\, z_i(x)[c]$$

$$\geq \sum_i \pi_i(x)\, z_i(x)[y] - \sum_i \pi_i(x)\, \max_{c \neq y} z_i(x)[c] = \sum_i \pi_i(x) m_i(x),$$

using $\max_c \sum_i a_{ic} \leq \sum_i \max_c a_{ic}$. With $m_{i^\star}(x) > 0$ and $m_j(x) \leq 0$ for $j \neq i^\star$,

$$\sum_i \pi_i(x) m_i(x) \;\geq\; \pi_{i^\star}(x)\, m_{i^\star}(x) + (1 - \pi_{i^\star}(x)) \min_{j \neq i^\star} m_j(x).$$

Thus $m_{\mathrm{mix}}(x) > 0$ whenever

$$\pi_{i^\star}(x)\, m_{i^\star}(x) + (1 - \pi_{i^\star}(x)) \min_{j \neq i^\star} m_j(x) > 0,$$

which rearranges to $\pi_{i^\star}(x) > \alpha$ as stated. $\square$

The previous lemma considered the binary-style case where only one expert is correct and all others are strictly incorrect. We now extend the analysis to the multiclass setting by allowing non-causal experts to have bounded negative margins. This produces a similar threshold condition on the gating weight, ensuring that the correct expert dominates when sufficiently favored by the gate.

**Lemma A.2** (Multiclass sparse threshold under bounded negative margins). *Let the one-vs-rest margin of expert $i$ at input $x$ be*

$$m_i(x) := z_i(x)[y] - \max_{c \neq y} z_i(x)[c].$$

*Suppose there exists a (causal) expert $i^\star$ and constants $m > 0$ and $\gamma \geq 0$ such that*

$$m_{i^\star}(x) \geq m, \qquad m_j(x) \geq -\gamma \text{ for all } j \neq i^\star.$$

*Then the mixture margin satisfies*

$$m_{\mathrm{mix}}(x) \geq \sum_{i=1}^{K} \pi_i(x)\, m_i(x) \geq \pi_{i^\star}(x)\, m - \big(1 - \pi_{i^\star}(x)\big)\gamma,$$

*and in particular $m_{\mathrm{mix}}(x) > 0$ whenever*

$$\pi_{i^\star}(x) > \frac{\gamma}{m + \gamma}.$$

*Proof.* By the max-sum inequality $\max_c \sum_i a_{ic} \leq \sum_i \max_c a_{ic}$,
$$m_{\mathrm{mix}}(x) = z_{\mathrm{mix}}(x)[y] - \max_{c \neq y} z_{\mathrm{mix}}(x)[c]$$

$$= \sum_i \pi_i(x)\, z_i(x)[y] - \max_{c \neq y} \sum_i \pi_i(x)\, z_i(x)[c] \geq \sum_i \pi_i(x)\, m_i(x).$$

Using the margin bounds $m_{i^\star}(x) \geq m$ and $m_j(x) \geq -\gamma$ for $j \neq i^\star$,

$$\sum_i \pi_i(x)\, m_i(x) \geq \pi_{i^\star}(x)\, m + \sum_{j \neq i^\star} \pi_j(x)\,(-\gamma) = \pi_{i^\star}(x)\, m - \big(1 - \pi_{i^\star}(x)\big)\gamma.$$

Hence
$$m_{\mathrm{mix}}(x) \geq \pi_{i^\star}(x)\, m - \big(1 - \pi_{i^\star}(x)\big)\gamma,$$

which is strictly positive precisely when $\pi_{i^\star}(x) > \gamma/(m + \gamma)$. $\qquad \square$

## A.2 Loss gap implies sparsity

The margin-based results above illustrate the role of sparsity in terms of logits and decision boundaries. We now provide a complementary perspective using losses directly. This proposition shows that a positive loss gap between the best expert and all others induces a lower bound on the gating weight assigned to the best expert. This connects the concept of expert specialization to loss-based analysis.

**Proposition A.3** (Loss gap implies sparsity). *Let $\{\ell_i(x,y)\}_{i=1}^{K}$ be the per-expert losses and $i^\star(x,y) \in \arg\min_{i \in [K]} \ell_i(x,y)$ be any minimizer. Define the mixture-of-losses $\bar{\ell}(x,y) = \sum_{i=1}^{K} \pi_i(x)\, \ell_i(x,y)$ and the loss gap*

$$\Delta(x,y) := \begin{cases} \min_{k \neq i^\star(x,y)} \big(\ell_k(x,y) - \ell_{i^\star}(x,y)\big), & K \geq 2, \\ 0, & K = 1. \end{cases}$$

*Then*
$$\bar{\ell}(x,y) \geq \ell_{i^\star}(x,y) + \big(1 - \pi_{i^\star}(x)\big)\Delta(x,y).$$

*Equivalently, for any $\Delta(x,y) > 0$,*

$$\pi_{i^\star}(x) \geq 1 - \frac{\bar{\ell}(x,y) - \ell_{i^\star}(x,y)}{\Delta(x,y)}.$$

*Proof.* Since $i^\star$ is a minimizer, for all $j \neq i^\star$ we have $\ell_j(x,y) - \ell_{i^\star}(x,y) \geq \Delta(x,y)$. Therefore,

$$\bar{\ell}(x,y) - \ell_{i^\star}(x,y) = \sum_{j=1}^{K} \pi_j(x)\big(\ell_j(x,y) - \ell_{i^\star}(x,y)\big) = \sum_{j \neq i^\star} \pi_j(x)\big(\ell_j(x,y) - \ell_{i^\star}(x,y)\big) \geq$$

$$\sum_{j \neq i^\star} \pi_j(x)\,\Delta(x,y) = \big(1 - \pi_{i^\star}(x)\big)\Delta(x,y),$$

which yields the stated inequality. Rearranging gives the equivalent lower bound on $\pi_{i^\star}(x)$. $\qquad \square$

## A.3  OOD RISK: COVERAGE + SELECTION

Finally, we return to the main theorem on OOD risk. This result decomposes the risk of the MoE into three interpretable terms: the oracle risk (if the aligned expert were always selected), a coverage penalty (arising when no expert perfectly matches the test distribution), and a selection penalty (arising when the gating function fails to concentrate on the aligned expert). The proof makes explicit the role of Jensen's inequality and shows how the sparsity results above fit into the broader risk bound.

**Theorem A.4** (OOD risk: coverage + selection). *Fix an OOD environment $\mathcal{D}' = \mathcal{D}_{m'}$. For the environment-aligned expert $i^\star(m')$, define*

$$\Gamma_{m'}(x,y) := \max_{j \neq i^\star(m')} \big(\ell_j(x,y) - \ell_{i^\star(m')}(x,y)\big).$$

*Then, under Assumptions 3.5 and 3.4,*

$$R_{\mathcal{D}'}(\mathrm{MoE}) \;\leq\; \underbrace{R_{\mathcal{D}'}(h^\star_{m'})}_{\textit{oracle risk}} + \underbrace{\varepsilon_{\mathrm{cov}}(m')}_{\textit{coverage via diversity}} + \underbrace{\mathbb{E}_{(x,y)\sim\mathcal{D}'}\big[(1 - \pi_{i^\star(m')}(x))\,\Gamma_{m'}(x,y)\big]}_{\textit{selection penalty via sparsity}}.$$

*Proof.* By convexity of cross-entropy in the logits,

$$R_{\mathcal{D}'}(\mathrm{MoE}) = \mathbb{E}_{\mathcal{D}'}\big[\ell_{\mathrm{CE}}(z_{\mathrm{mix}}(x),y)\big] \;\leq\; \mathbb{E}_{\mathcal{D}'}\big[\bar{\ell}(x,y)\big], \qquad \bar{\ell}(x,y) := \sum_{i=1}^{K} \pi_i(x)\,\ell_i(x,y).$$

Fix $(x,y)$ and abbreviate $i^\star = i^\star(m')$. Decompose

$$\bar{\ell}(x,y) = \pi_{i^\star}(x)\,\ell_{i^\star}(x,y) + \sum_{j \neq i^\star} \pi_j(x)\,\ell_j(x,y) = \ell_{i^\star}(x,y) + \sum_{j \neq i^\star} \pi_j(x)\big(\ell_j(x,y) - \ell_{i^\star}(x,y)\big).$$

By definition of $\Gamma_{m'}(x,y)$, each difference satisfies $\ell_j(x,y) - \ell_{i^\star}(x,y) \leq \Gamma_{m'}(x,y)$, hence

$$\bar{\ell}(x,y) \;\leq\; \ell_{i^\star}(x,y) + \Big(\sum_{j \neq i^\star} \pi_j(x)\Big)\Gamma_{m'}(x,y) = \ell_{i^\star}(x,y) + \big(1 - \pi_{i^\star}(x)\big)\Gamma_{m'}(x,y).$$

Taking expectations under $\mathcal{D}'$ yields

$$R_{\mathcal{D}'}(\mathrm{MoE}) \;\leq\; \mathbb{E}_{\mathcal{D}'}\big[\ell_{i^\star}(x,y)\big] + \mathbb{E}_{\mathcal{D}'}\big[(1 - \pi_{i^\star}(x))\,\Gamma_{m'}(x,y)\big] =$$

$$R_{\mathcal{D}'}(h_{i^\star}) + \mathbb{E}_{\mathcal{D}'}\big[(1 - \pi_{i^\star}(x))\,\Gamma_{m'}(x,y)\big].$$

Finally, by Assumption 3.5, $R_{\mathcal{D}'}(h_{i^\star}) \leq R_{\mathcal{D}'}(h^\star_{m'}) + \varepsilon_{\mathrm{cov}}(m')$, which proves the claim. □

## B  DETAILED RELATED WORKS

### B.1  CAUSAL SUBGRAPH-BASED OOD LEARNING.

LECI enforces label–environment independence by learning edge masks such that the label is independent of the environment and the environment is independent of the causal subgraph (Gui et al., 2023). UIL jointly enforces semantic and structural invariance, aligning graphs across environments using graphon distances (Sui et al., 2025). LIRS instead learns spurious features first and removes them from ERM-learned features, thereby capturing a broader set of invariant subgraphs (Yao et al., 2025). DIR discovers label-causal subgraph rationales by intervening on the training distribution and selecting features invariant across the induced environments, filtering out spurious shortcuts (Wu et al., 2022). CIGA learns causally invariant graph representations by extracting subgraphs that maximally preserve label-relevant intra-class information (Chen et al., 2022). GSAT applies an information-bottleneck–driven stochastic attention to mask task-irrelevant nodes/edges, yielding faithful rationales (Miao et al., 2022). CSIB learns label-causal subgraphs under a causal model while jointly optimizing invariant risk with a graph information bottleneck to balance invariance versus compression for FIIF/PIIF (An et al., 2024).

## B.2 INSTANCE HETEROGENEITY.

GSAT enforces adversarial consistency to discourage reliance on unstable features, thereby enabling more robust predictions across heterogeneous samples (Miao et al., 2022). GALA explicitly models multiple latent graph views and aggregates them to capture sample-level causal diversity (Chen et al., 2023). AIA generates new environments through adversarial augmentation while preserving stable features, simulating heterogeneity under covariate shift (Sui et al., 2023). Other approaches such as FLAG (Lu et al., 2024), StableGNN (Yao et al., 2024), and GraphMETRO (Wu et al., 2024) similarly expand training coverage via data augmentation.

GSAT enforces adversarial consistency to discourage reliance on unstable features, thereby enabling more robust predictions across heterogeneous samples (Miao et al., 2022). GALA explicitly models multiple latent graph views and aggregates them to capture sample-level causal diversity (Chen et al., 2023). AIA generates new environments through adversarial augmentation while preserving stable features, simulating heterogeneity under covariate shift (Sui et al., 2023). FLAG enforces large-scale augmentation by applying instant feature-level adversarial noise during training while preserving graph topology (Lu et al., 2024). StableGNN pools subgraphs into high-level variables and adds a causal-variable distinguishing penalty term to prioritize stable correlations under distribution shifts (Fan et al., 2023).

## B.3 MIXTURE-OF-EXPERTS FOR GNNS.

MoE architectures allocate computation across experts selected by a gating function (Jacobs et al., 1991; Shazeer et al., 2017). On graphs, GMoE leverages MoE layers to scale graph transformers efficiently (Wang et al., 2023a). MixGNN employs expert routing to improve efficiency on large-scale graph tasks (Chen et al., 2025). Other variants (Hu et al., 2022) similarly focus on distributing computation or scaling to large graphs. These methods demonstrate the potential of MoE for graphs but do not target OOD generalization.

## B.4 GRAPHMETRO.

In the OOD setting, GraphMETRO decomposes distributional heterogeneity and aligns referential representations across shifts by a gated mixture-of-experts with shift-specialized experts (Wu et al., 2024). However, this design requires the selection of shift types before training with no guarantee that the selected shift types are the ones that will be encountered at test time. Furthermore, these perturbations risk altering label semantics unknowingly. In contrast, our approach uses MoE not for augmentation, but for *causal subgraph identification*, encouraging experts to extract diverse causal subgraphs and thereby directly modeling instance heterogeneity.

## B.5 OUR APPROACH.

Unlike prior methods, we do not rely on strong assumptions about the underlying SCM, as in many causal approaches, nor do we risk altering the label semantics through perturbations, as in augmentation-based methods. Our framework instead learns to identify candidate causal subgraphs through expert-specific masks, with a sparse gating mechanism selecting among them on a per-instance basis. This design encourages experts to specialize in distinct causal explanations and allows the model to adaptively choose the most relevant one for each input, thereby capturing instance-level heterogeneity.

# C  IMPLEMENTATION DETAILS

## C.1 EXPERT SUBGRAPH EXTRACTION

Given an input graph $x = (V, E, X)$, we first compute node embeddings using a shared GNN encoder. Each expert $k \in [K]$ produces *edge-level mask logits* $\ell_e^{(k)}$ via a small expert-specific MLP applied to the concatenated embeddings of the edge endpoints. The logits are transformed into binary masks using a Gumbel–sigmoid straight-through estimator:

$$m_e^{(k)} = \mathbf{1}\left\{\sigma\left(\tfrac{\ell_e^{(k)}+g}{\tau}\right) > 0.5\right\},$$

where $g$ is Gumbel noise, $\sigma$ is the sigmoid, and $\tau = 0.1$ is the temperature. Hard binary masks $m_e^{(k)}$ are used in the forward pass to produce the masked graph $G^{(k)}$, while gradients flow through the continuous relaxation during backpropagation. Node weights are induced from their incident edges, and isolated nodes are removed. Each expert has its own GNN encoder and classifier head. Unless otherwise specified, we adopt the Graph Isomorphism Network (GIN) following the GOOD benchmark configuration: hidden dimension 300, depth 3, dropout 0.5.

## C.2   TASK LOSS

For each expert $k$, we compute the per-sample cross-entropy

$$\ell_{\text{CE}}^{(k)}(x, y) = \text{CE}\big(\theta_k(G^{(k)}), y\big).$$

The overall task loss is a gate-weighted aggregation across experts:

$$\ell_{\text{CE}}(x, y) = \sum_{k=1}^{K} \pi_k(x)\, \ell_{\text{CE}}^{(k)}(x, y),$$

ensuring that experts favored by the gate receive stronger gradients, thereby encouraging specialization.

## C.3   MASK REGULARIZATION

To control the size of extracted subgraphs, we regularize the average fraction of edges retained by each expert. For expert $k$, the observed keep-rate on graph $g$ is

$$\hat{\rho}_g^{(k)} = \frac{1}{|E|} \sum_{e \in E} m_e^{(k)}.$$

We penalize deviations from a target $\rho \in [0, 1]$ via

$$\ell_{\text{reg}}^{(k)} = \big(\hat{\rho}_g^{(k)} - \rho\big)^2.$$

This discourages degenerate solutions where experts keep either too few or too many edges.

## C.4   DIVERSITY LOSS

To prevent collapse of experts onto identical subgraphs, we enforce the semantic diversity condition (Definition 3.1). Masks are standardized, correlations are computed, and we penalize high off-diagonal correlations:

$$\ell_{\text{div}} = \frac{1}{K(K-1)} \sum_{i \neq j} \max\{0,\ |C_{ij}^{(g)}| - \tau_{\text{corr}}\},$$

where $C_{ij}^{(g)}$ is the correlation between standardized masks of experts $i$ and $j$ on graph $g$. This loss directly encourages experts to specialize on different subgraphs.

## C.5   GATING MECHANISM

**Gate inputs.**   The gate does not operate directly on the raw graph but instead consumes diagnostic, label-free features derived from each expert's outputs. For expert $k$ and sample $b$, we construct a feature vector $\Phi_{b,k} \in \mathbb{R}^{10}$:

$$\Phi_{b,k} = \Big[ \max_c p_{b,k}(c),\ \text{margin}_{b,k},\ H(p_{b,k}),\ -\log \sum_c e^{z_{b,k}(c)},\ -\text{KL}(p_{b,k} \,\|\, p_{b,k}^{\text{weak}}),$$

$$-\frac{1}{K-1} \sum_{j \neq k} \text{KL}(p_{b,k} \,\|\, p_{b,j}),\ H(p_{b,k}^{\text{env}}),\ H(p_{b,k}^{\text{spur}}),\ n_b,\ m_b \Big],$$

where $p_{b,k} = \text{softmax}(z_{b,k})$ are class probabilities from expert $k$, $\text{margin}_{b,k}$ is the difference between the top-1 and top-2 probabilities, $H(\cdot)$ is Shannon entropy, and the energy term is

$-\log \sum_c \exp(z_{b,k}(c))$. The KL terms capture stability under weak augmentations and pairwise disagreement between experts. $p_{b,k}^{\text{env}}$ and $p_{b,k}^{\text{spur}}$ are outputs of environment and spurious classifiers, while $n_b$ and $m_b$ denote the number of nodes and edges. The feature vector is passed through a two-layer MLP with hidden dimension 64 and ReLU activations. Outputs are normalized with Entmax($\alpha = 1.38$), yielding sparse gate probabilities $\pi_{b,1:K}$.

**Gate loss.** The gate is trained with three complementary objectives:

(i) *Teacher–student alignment.* A teacher distribution $q_b$ is constructed from expert competence signals:

$$r_{b,k} = -\ell_{\text{CE}}(z_{b,k}, y_b) - w_{\text{la}}\, \ell_{\text{LA},b,k} - w_{\text{ea}}\, \ell_{\text{EA},b,k}, \quad q_b = \text{softmax}\left(\frac{r_b}{\tau_{\text{oracle}}}\right).$$

The student is the gate output $p_b = \text{Entmax}_{\alpha=1.38}(s_b)$, where $s_b$ are gate scores. The alignment loss is

$$L_{\text{align}} = \text{KL}(q_b \,\|\, p_b).$$

(ii) *Balanced usage.* To avoid collapse to a single expert, we regularize the average gate distribution over a batch:

$$L_{\text{balance}} = \text{KL}(u \,\|\, \bar{p}), \qquad \bar{p} = \tfrac{1}{B}\sum_b p_b,$$

where $u$ is uniform over experts.

(iii) *Per-sample sparsity.* To encourage sparse routing, we penalize high-entropy gate outputs:

$$L_{\text{sparse}} = \tfrac{1}{B}\sum_{b=1}^{B} H(p_b), \qquad H(p_b) = -\sum_{k=1}^{K} p_{b,k} \log p_{b,k}.$$

The final gate loss is

$$L_{\text{gate}} = L_{\text{align}} + L_{\text{balance}} + L_{\text{sparse}}.$$

## C.6 FINAL OBJECTIVE

The overall training loss is

$$\mathcal{L} = \ell_{\text{CE}} + \lambda_{\text{reg}}\ell_{\text{reg}} + \lambda_{\text{div}}\ell_{\text{div}} + \lambda_{\text{gate}}L_{\text{gate}}.$$

Here $\ell_{\text{reg}} = \frac{1}{K}\sum_k \ell_{\text{reg}}^{(k)}$, and $\lambda_{\text{reg}}, \lambda_{\text{div}}, \lambda_{\text{gate}}$ are loss weights. This formulation integrates the task, coverage, selection, and diversity principles while guarding against degenerate solutions and expert starvation.

## C.7 HYPERPARAMETER SETTINGS

For all experiments, we adopt a common set of fixed hyperparameters. Each expert network is implemented as a Graph Isomorphism Network (GIN) with hidden dimension 300, three layers, and a dropout rate of 0.5. The gating network is a two-layer MLP with hidden dimension 64 and ReLU activations, followed by an Entmax transformation with $\alpha = 1.38$ to produce sparse routing probabilities. The loss weights are set to 1.0 for the task cross-entropy, mask regularization, and diversity losses, and 0.1 for the gate loss. Optimization is performed with Adam using a weight decay of $10^{-4}$. The learning rate is reduced by a factor of two if the validation performance does not improve by at least 0.001 for 10 consecutive epochs.

In addition to these fixed settings, we tune a small number of hyperparameters. The mask keep-rate prior $\rho$ is sampled uniformly between 0.1 and 0.9. The learning rate is selected from $\{0.001, 0.0005, 0.0001\}$, and the batch size is chosen from $\{32, 64, 128, 256, 512\}$. Hyperparameter tuning is conducted with 10 independent trials per dataset, each initialized with a different random seed. The best configuration is selected based on validation performance, retrained on the training set, and finally evaluated on the held-out test sets over five different seeds. This procedure ensures consistent model selection without any test leakage.

# D    CAUSAL ASSUMPTIONS

**Problem setup.**    Each input is a graph $x = (V, E, X)$ with label $y \in \mathcal{Y}$. Let $G_c$ denote *causal/stable* substructures and $G_s$ denote *spurious/environmental* substructures. We focus on *co-variate shift*: the marginal over graphs changes across environments, while $p(y \mid x)$ is stable, i.e., $p_{\text{train}}(x) \neq p_{\text{test}}(x)$ but $p_{\text{train}}(y \mid x) = p_{\text{test}}(y \mid x)$.

**FIIF vs. PIIF.**    Following (Gui et al., 2023), fully/partially informative invariant features formalize how $G_s$ interacts with $G_c$ and environments $E$. Under **FIIF**, $G_s$ is influenced by both $G_c$ and $E$, creating spurious dependence $Y \leftarrow G_c \rightarrow G_s$; under **PIIF**, a collider induces $G_s \leftrightarrow Y$ through $G_c$ (e.g., $Y \leftarrow G_c \rightarrow G \leftarrow G_s$). These regimes explain why $G_s$ can correlate with $Y$ even when noncausal.

WHAT PRIOR METHODS ASSUME

**LECI (Gui et al., 2023).**    LECI jointly enforces (i) *label–environment causal independence*: $G_c \perp E$ and (ii) *label–spurious independence*: $G_s \perp Y$ (operationalized via adversarial objectives), aiming to recover invariant subgraphs. In FIIF/PIIF regimes, these constraints are used to filter $G_s$ and keep $G_c$.

**UIL (Sui et al., 2025).**    UIL posits a stronger *structural* invariance: the stable (causal) part of graphs within each class shares a class-specific *graphon* pattern that is invariant across all (seen/unseen) environments; semantic invariance is then layered on top. In effect, $G_c$ is assumed *structurally invariant across environments*.

**DIR and related.**    DIR (Wu et al., 2022) also targets invariant rationales/causal attention, typically constructing interventional/augmented environments and enforcing invariance of the predictive mechanism across them.

WHY THESE ASSUMPTIONS BREAK IN PRACTICE

**Concrete examples.**

- **Chemistry.** Multiple active chemotypes for the same endpoint; scaffold/time/source shifts alter $G_s$ and even the prevalence of certain $G_c$'s, violating structural uniqueness and $G_s \perp Y$ (Wu et al., 2018; Hu et al., 2020).
- **Social/text graphs.** Domain-specific syntax/community structures change across splits; $G_s$ (e.g., degree/length) correlates with $Y$ via FIIF/PIIF pathways (Gui et al., 2022).

**General reasons.** (i) *Instance heterogeneity*: real tasks often admit *multiple* causal explanations within the same class (different $G_c$'s per instance). In molecular property prediction, distinct chemotypes/functional groups can yield the same label (e.g., multiple acidic moieties), and scaffold splits explicitly emphasize cross-chemotype variation; thus any single class-graphon assumption can be violated. (ii) *$Y$–$G_s$ correlation (FIIF/PIIF)*: even when $G_s$ is noncausal, it may correlate with $Y$ via $G_c$, making $G_s \not\perp Y$ and breaking LECI-style independence assumptions. Recent theory shows that environment augmentation cannot, in general, identify invariance without additional biases; moreover, $G_s$ and $Y$ can have *arbitrary* correlation, making environment inference/labeling fundamentally hard (Chen et al., 2023).

OUR ASSUMPTION (MINIMAL AND ROBUST)

We assume only that **each graph admits at least one causal subgraph $G_c$ that governs $Y$**. We do *not* assume (i) $G_c$ is unique within a class, (ii) $G_c$ is structurally identical across environments (no class-graphon), or (iii) $G_s \perp Y$ or $G_c \perp E$. This minimal assumption tolerates instance-level heterogeneity (different $G_c$'s per instance), admits FIIF/PIIF couplings, and aligns with practical datasets where multiple mechanisms yield the same label (e.g., multiple binding motifs or syntax patterns). **Benefit:** we avoid brittle structural/independence assumptions and instead learn to *select* among diverse causal hypotheses at the instance level.

Table 7: Performance of the MoE framework with 1, 4, and 8 experts. Standard deviations are reported in parentheses.

| Experts | HIV ↑ | | Twitter ↑ | Motif ↑ | | SST2 ↑ | Avrg ↑ |
|---|---|---|---|---|---|---|---|
| | Scaffold | Size | Length | Basis | Size | Length | |
| 1 | 67.13 (2.2) | 63.48 (0.9) | 58.68 (1.7) | 89.3 (2.1) | 65.31 (1.2) | 81.11 (1.1) | 70.84 |
| 4 | 70.65 (1.3) | 65.35 (1.3) | 60.48 (0.7) | 91.79 (1.5) | 76.40 (2.1) | 83.41 (1.2) | 74.68 |
| 8 | 71.55 (1.4) | 66.98 (1.0) | 61.13 (1.1) | 92.8 (1.4) | 75.52 (2.9) | 83.73 (1.4) | 75.29 |

EMPIRICAL SUPPORT: SINGLE-EXPERT PERFORMANCE

Table 7 provides empirical evidence for the strength of this assumption-light design. In the single-expert setting, the model reduces to the simplest form of causal subgraph extraction: a single extractor, regularized by $\rho$, feeding into a GNN trained like standard ERM. The GNN is optimized only with the task loss, while the extractor is trained with the task loss plus the regularization term. This design makes no additional assumptions about the underlying causal SCM beyond the sparsity prior, representing a minimal instantiation of causal subgraph methods. Remarkably, despite its simplicity, this approach outperforms all but one causal baseline (LECI) in average performance across all datasets. This result suggests that the explicit structural assumptions encoded in prior causal methods may in fact be too restrictive or fragile, and that a more assumption-light approach can provide stronger and more reliable generalization.

# E  DATASET DETAILS

We provide dataset-specific information for the four GOOD tasks used in the main paper. Each task introduces a different type of structural or distributional shift, following the benchmark design in Gui et al. (2022).

**GOOD-HIV.**  This task is a molecular property prediction problem derived from the MoleculeNet HIV dataset. Each graph corresponds to a molecule, where nodes are atoms and edges are chemical bonds. The prediction task is binary classification: whether the molecule inhibits HIV replication. To evaluate OOD generalization, two types of environment splits are defined. The *scaffold split* partitions molecules according to their core scaffolds, such that training and test sets contain molecules with distinct underlying structures. The *size split* creates a distribution shift by separating molecules based on the number of heavy atoms, exposing models to molecules of substantially different sizes at test time. These shifts test whether models can generalize beyond memorized molecular backbones and size ranges.

**GOOD-Motif.**  This synthetic dataset is designed to provide controlled graph-level classification tasks with interpretable shifts. Each graph is generated by attaching motifs (e.g., cycles, cliques, houses) onto a random base graph. The label depends on the presence or type of motif. The benchmark defines two kinds of shifts. In the *motif-basis split*, the set of motifs used for training differs from those used in evaluation, requiring extrapolation across structural primitives. In the *size split*, the base graphs differ in size between environments, requiring robustness to distribution shifts in graph order and density. This dataset isolates structural shifts in a controlled synthetic setting, making it useful for probing whether models can truly capture causal motif information.

**GOOD-Twitter.**  This dataset consists of ego-networks from Twitter users. Each ego-network is represented as a graph where the central node corresponds to the ego user, and edges represent social connections among the ego and their neighbors. The task is binary classification of user attributes. The primary distribution shifts are *domain shifts* across different user communities, which result in differences in graph sparsity, degree distributions, and local structural motifs. Since ego-networks are sampled from diverse domains, training and test sets differ significantly in their structural properties, requiring models to generalize across heterogeneous social network subgraphs.

**GOOD-SST2.**  This dataset is based on the Stanford Sentiment Treebank 2 (SST2). Each sentence is parsed into a dependency tree, which serves as the input graph. The task is binary classification of

sentence sentiment (positive vs. negative). The distribution shifts are introduced by partitioning the data according to linguistic structures. Specifically, environments differ in the average tree depth and branching factors, leading to structural shifts in dependency graphs. These shifts test whether graph models can capture sentiment cues in syntactic structures when faced with substantial variation in parse tree topology across domains.

Overall, these four datasets cover both *real-world* domains and *synthetic* graphs, and they introduce diverse OOD challenges, including scaffold and size shifts, motif-basis changes, domain heterogeneity, and structural variation. This variety makes them a comprehensive testbed for evaluating the robustness of graph OOD methods.

# F INTERPRETABILITY RESULTS

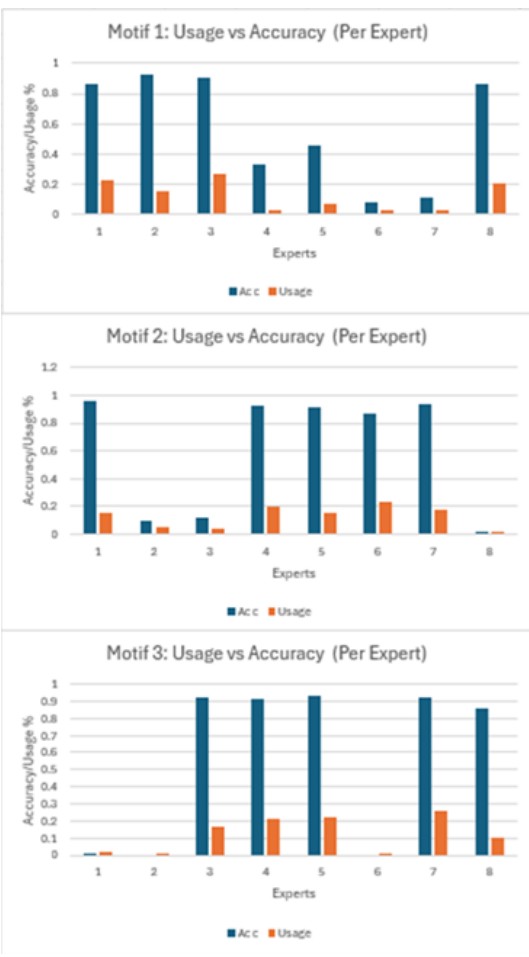

Figure 2: Per-expert usage and accuracy across motifs (MOTIF-BASIS). For each motif, we report the average gating probability (*usage*) and the corresponding per-expert accuracy, averaged over all test instances containing that motif.

This section provides an interpretable diagnostic showing how the gating network routes instances exhibiting different heterogeneous causal subgraphs to different experts, and whether routing aligns with expert performance.

We use the MOTIF-BASIS dataset from the GOOD benchmark, which contains three heterogeneous causal subgraphs (three distinct motifs) (Gui et al., 2022). After training, we evaluate on the test set and compute, for each motif $m \in \{1, 2, 3\}$ and each expert $k \in \{1, \dots, K\}$ (here $K = 8$): (i) **usage**, defined as the average gating probability assigned to expert $k$, averaged over all test instances

containing motif $m$; and (ii) **accuracy**, defined as the average classification accuracy of expert $k$ on the same subset of instances. This yields $3 \times 8$ (usage, accuracy) pairs summarizing routing behavior and expert specialization.

### F.1    PER-MOTIF ROUTING PATTERNS.

Figure 2 visualizes per-expert usage for each motif. We can observe that for each motif, the gate concentrates probability mass on a subset of experts rather than using all experts uniformly. Furthermore, the subset of frequently-used experts changes across motifs, indicating that the gate responds to the underlying causal subgraph. Concretely, motif 1 primarily uses experts $\{1, 2, 3, 8\}$; motif 2 primarily uses $\{1, 4, 5, 6, 7\}$; and motif 3 primarily uses $\{3, 4, 5, 7, 8\}$. Several experts are nearly unused for specific motifs (e.g., expert 6 for motifs 1 and 3), while becoming prominent for others (e.g., expert 6 for motif 2). This demonstrates that the experts are indeed trained to specialize on different causal subgraphs, empirically supporting our claim of semantic diversity.

### F.2    ALIGNMENT BETWEEN ROUTING SELECTION AND EXPERT PERFORMANCE.

Figure 3 plots, across all $(m, k)$ pairs, expert usage vs. expert accuracy, with points grouped by motif category. Routing aligns with expertise: experts that are more frequently selected for a motif tend to achieve higher accuracy on instances containing that motif. Quantitatively, a Pearson correlation test between usage and accuracy over all $(m, k)$ pairs yields $r = 0.823$ with $p < 10^{-7}$, indicating a strong positive association between gate probability and expert correctness for the corresponding causal subgraph which supports our claim of correct expert selection.

### F.3    VISUALIZATION OF PREDICTED SUBGRAPHS.

Figure 4 visualizes the subgraphs extracted by a four-expert MoE on three training-set graphs from Motif-basis containing the house motif. These visualizations show that different experts extract distinct subgraphs, demonstrating semantic diversity. Moreover, for each instance, only one or two experts correctly recover the house motif, highlighting expert specialization.

Overall, these results provides strong evidence for the interpretability of our method and visually support our claims of diverse expert specialization and successful routing alignment.

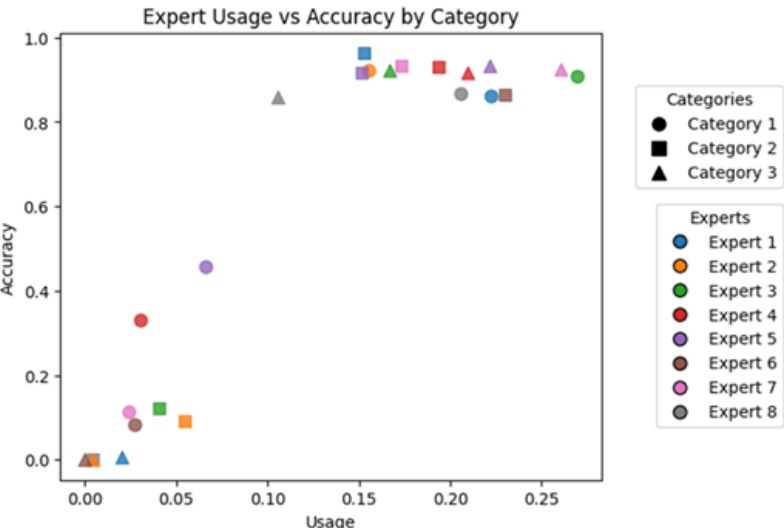

Figure 3: Usage vs. accuracy by motif category. Each point corresponds to one (motif, expert) pair. Higher usage aligns with higher accuracy, suggesting that the gate routes instances toward experts that specialize in the corresponding motif.

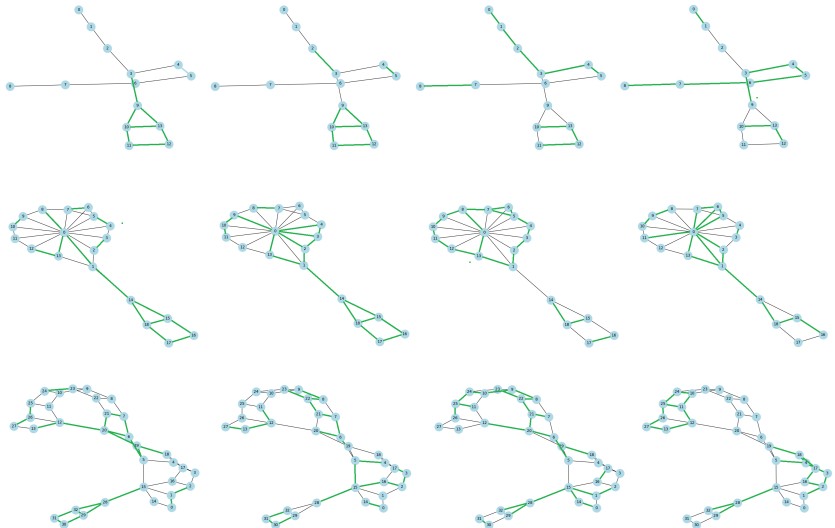

Figure 4: Visualization of the subgraphs extracted by a four-expert MoE on three training-set graphs from Motif-basis containing the house motif. Edges highlighted in green are extract by the expert.

Table 8: Effect of learned gating on Twitter and SST2. Uniform: uniform averaging of logits (no gating). Majority: majority voting (no gating). Top 1: sparse selection of the top-1 expert predicted by the gate. Top 2: sparse selection of the top-2 experts predicted by the gate. Gate Weight (Soft): weighted averaging of logits using the gate weights (used in our method).

| Learned Gating | Twitter | SST2 |
|---|---|---|
| Uniform | 59.92 | 82.72 |
| Majority | 58.39 | 82.99 |
| Top 1 | 56.36 | 79.36 |
| Top 2 | 59.11 | 82.13 |
| Gate Weight (Soft) | 61.13 | 83.73 |

# G    ADDITIONAL EXPERIMENTS

In this section, we provide additional experiments and discussions.

## G.1    HIV DATASET

For HIV dataset, we apply class balanced sampling during batch sampling and logit adjustment during training (Wang et al., 2023b). This follows the convention of LIRS and GALA who apply similar techniques to mitigate the class imbalance (Yao et al., 2025; Chen et al., 2023). Without these techniques, our performance on HIV-Scaffold and HIV-Size drops slightly from 71.55 to 70.65 and 66.98 to 65.35, respectively. In this case, our method still achieves the best average performance and average rank across all evaluated baselines.

## G.2    GATING MECHANISM

We next assess the contribution of the gating mechanism. Our framework employs a learned gate that assigns input-dependent weights to experts, forming a weighted combination of their predictions. Table 8 compares this design to several alternatives. On both Twitter and SST2, uniform averaging already benefits from expert complementarity, but it falls short of the learned gate (59.92 vs. 61.13 on Twitter, 82.72 vs. 83.73 on SST2). Majority voting performs similarly or worse (58.39 and 82.99), suggesting that ignoring the confidence of individual experts limits robustness. Sparse selection of only the top-1 or top-2 experts consistently underperforms (e.g., 56.36/79.36 and 59.11/82.13),

Table 9: Computational efficiency analysis by varying number of experts (1, 4, 8, 16, 32) and measuring training time, inference time, and OOD accuracy on Twitter, SST2, and Motif-Basis.

|  | Num Experts | Training Time | Inference Time | Parameters | Accuracy |
|---|---|---|---|---|---|
| Twitter | 1 | 3m12s | 4s | 328,641 | 58.68 |
|  | 4 | 3m18s | 4s | 11,091,324 | 60.48 |
|  | 8 | 3m30s | 4s | 21,497,876 | **61.13** |
|  | 16 | 3m47s | 5s | 42,310,980 | 60.05 |
|  | 32 | 4m34s | 6s | 83,937,188 | 60.12 |
| SST2 | 1 | 38m35s | 26s | 3,284,002 | 81.11 |
|  | 4 | 45m19s | 42s | 11,081,692 | 83.41 |
|  | 8 | 58m47s | 50s | 21,478,612 | 83.73 |
|  | 16 | 1h15m21s | 50s | 42,272,452 | 83.72 |
|  | 32 | 1h56m05s | 1m20s | 83,860,132 | **83.92** |
| Motif-Basis | 1 | 13m45s | 6s | 2,361,796 | 83.9 |
|  | 4 | 15m38s | 6s | 8,083,168 | 91.79 |
|  | 8 | 17m07s | 6s | 15,711,664 | **92.8** |
|  | 16 | 22m34s | 10s | 30,968,656 | 92.47 |
|  | 32 | 31m16s | 12s | 61,482,640 | 92.33 |

showing that over-reliance on top-k experts on all samples discards useful complementary information. By contrast, the learned gate yields the best performance on both datasets, highlighting that adaptive weighting is crucial for fully exploiting expert specialization. Importantly, the gate not only aggregates predictions effectively but also mitigates expert collapse by encouraging specialization into complementary substructures.

### G.3 EMPIRICAL COMPUTATIONAL EFFICIENCY ANALYSIS

In Section 3.5, it was demonstrated that the computattional overhead grows linearly with respect to the number of experts. This is validated by the empirical results in Table 9. Notably, in practice, both training and inference times increase sub-linearly with the number of experts while OOD accuracy remains largely stable with slight improvements/degradations depending on the dataset. Furthermore, the results indicate that eight experts provides a good balance between computational efficiency and OOD accuracy.

### G.4 TRAINING STABILITY

Across all experiments, we did not observe training instability or divergence. To illustrate this, we report the training loss curves on MOTIF-BASIS in Figure 5 which shows that each component of the loss decreases smoothly and consistently. We attribute the observed stability to two design choices:

- Uniform gating warm-up: we employ uniform gating in the first $\ell = 10$ epochs to ensure that all experts receive sufficient training signal before the gating network begins selective routing. This warm-up ensures each expert learns a reasonable initial representation, which stabilizes early optimization.
- Module-specific loss: Different components of the loss optimize different modules of the model which reduces potential gradient conflicts. The gating module only receives the gating loss, the experts only receive the task loss (cross-entropy), the subgraph extractor is the only component that receives multiple losses (task, diversity, and regularization).

### LARGE LANGUAGE MODELS STATEMENT

Large Language Models were used exclusively for editorial purposes, such as refining language and improving readability. All scientific contributions were developed solely by the authors.

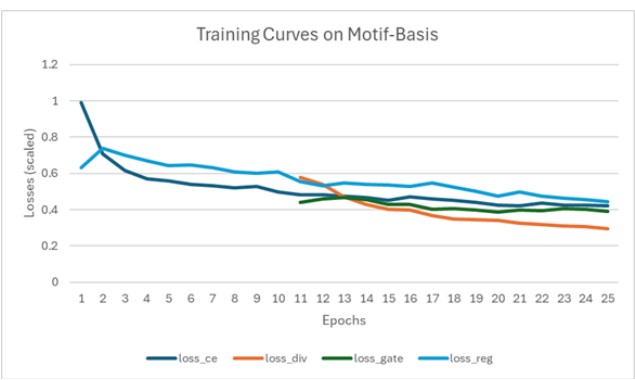

Figure 5: Training losses on MOTIF-BASIS. We use uniform gating for the first ten epochs; thus, the diversity and gating losses are reported starting at epoch 10.

