# OpenReview forum: "Diverse and Sparse Mixture-of-Experts for Causal Subgraph–Based Out-of-Distribution Graph Learning"
_ICLR.cc/2026/Conference — ICLR 2026 Poster_

### Official Review · Reviewer_nffD · 2025-10-18

**Soundness:** 3
**Presentation:** 3
**Contribution:** 3
**Rating:** 6
**Confidence:** 3

**Summary:**

This paper proposes a Mixture-of-Experts (MoE) framework to address the out-of-distribution (OOD) generalization problem for graphs. The method explicitly models the instance-level causal heterogeneity prevalent in real-world data by forcing experts to extract diverse causal subgraph hypotheses and using a sparse gating network to adaptively select the best explanation for each instance.

**Strengths:**

- S1: The paper is well-motivated, with a clear and easy-to-follow logic.

- S2: The paper provides a clear theoretical framework that decomposes the OOD risk into coverage and selection terms. It directly links the proposed diversity and sparsity mechanisms to the goal of reducing OOD error, effectively supporting the paper's core claims and contributions.

- S3: The authors have conducted extensive experiments on standard benchmarks to demonstrate the effectiveness of their method. Furthermore, the ablation studies validate the necessity of each core component of the framework.

**Weaknesses:**

- W1: The core premises of the paper, namely instance-level heterogeneity and the resulting causal diversity, require more formal and clear definitions and introductions.

- W2: Assumption 3.5 appears to be a critical condition for the method to achieve OOD generalization. It states that at least one expert can achieve satisfactory performance in the test environment. This assumption may be too strong, as it implicitly requires that the causal subgraph structures present in the test data must have been seen during training and learned by at least one expert.

- W3: The method relies on training multiple expert networks, yet it lacks a corresponding analysis of computational efficiency. Although the paper claims the method is scalable, it does not provide a quantitative analysis of the relationship between computational cost (e.g., training time) and the number of experts, K.

**Questions:**

- Q1 (Regarding W1): The concepts of instance-level heterogeneity and causal diversity need further elaboration. What is causal diversity, and is it a concept specific to the instance level? A related issue is the assumption of instance-level heterogeneity, which should be explicitly stated. Does this assumption imply that different instances can have different causal graphs? If so, why would different instances from the same data source follow different causal graph structures, and what are the corresponding real-world scenarios?

- Q2 (Regarding W2): My understanding of the method is as follows: different experts are used to extract different subgraph structures based on a diversity constraint. A selection mechanism then assigns the maximum predictive weight to the expert with the lowest prediction loss, and the subgraph extracted by this expert is considered the causal part. Is this understanding correct? If so, the generalization guarantee provided by Assumption 3.5 is, in essence, a strong constraint on the structure of the test graphs. Could you please discuss this?

---

> ### Author Response · Authors · 2025-11-19
> **Response to Reviewer nffD (Part 1)**
>
> We thank the reviewer for their insightful comments. We address each concern in the following sections and hope these clarifications resolve the reviewer’s questions.
>
> ---
>
> **W1\Q1: Instance-level Heterogeneity and Causal Diversity**
>
> > *Does this assumption imply that different instances can have different causal graphs?*
>
> Yes — the reviewer’s understanding is correct, and we appreciate the opportunity to clarify this more explicitly.
>
> As we state in Section 2 (Background):
>
> > *Instance heterogeneity means that the causal subgraph varies not only across environments but also across individual samples. In molecular property prediction, for example, one compound’s activity may depend on a particular functional group, while another’s relies on a different one (Wu et al., 2023; Inae et al., 2024; Wu et al., 2018).*
>
> In our framework:
>
> - **Instance-level heterogeneity** means that *even within the same data source, environment, and label*, different graph instances may rely on **different causal subgraphs** to determine their outcome.
> - **Causal diversity** refers to the **variation in these causal subgraphs across instances**. Formally, if each graph $G$ has a (possibly latent) causal subgraph $G_c$ that is sufficient to predict $Y$, then the collection of $G_c$'s need not be identical across instances; this variability is what we call causal diversity.
>
>
> Below we provide concrete real-world scenarios (aligned with the GOOD datasets we use) where this instance-level causal diversity naturally arises.
>
> - **GOOD-HIV (molecular property prediction).**
>   The label (“inhibits HIV replication”) can be achieved by **multiple chemotypes**:
>   - Some active molecules are **protease-inhibitor-like** (peptidomimetic backbones with specific secondary alcohol/amide patterns),
>   - Others resemble **reverse-transcriptase inhibitors** (heteroaromatic cores with particular hydrogen-bonding patterns),
>   - Others may exploit other binding modes.
>
>   In our terms, these different chemotypes correspond to **different causal subgraphs** (different $G_c$) that all yield the *same* label. Thus, two molecules in the *same dataset and environment*, both labeled “active”, can legitimately rely on different causal subgraphs: one may be active *because* of a protease-inhibitor motif, another *because* of an NNRTI-like heteroaromatic pattern.
>
>
> - **GOOD-SST2 (sentiment on dependency trees).**
>   Positive sentiment can be expressed through very different syntactic constructions:
>   - “This movie was *absolutely fantastic*” (direct positive adjective attached to the noun),
>   - “Not a bad film at all” (negation + mild negative word),
>   - “It *surprised me in a good way*” (verb phrase + subordinate clause).
>
>
>   On the dependency tree, these correspond to **different subtrees** (different $G_c$) being causally responsible for the same positive label. Again, the dataset and label are the same, but the *causal subgraph structure* that carries sentiment varies from sentence to sentence.
>
> In summary, **yes**, our assumption explicitly allows that **different instances from the same data source can rely on different causal subgraphs**, even for the same label and environment. This is exactly what we call *instance-level causal diversity*: real-world data often admit **multiple valid causal subgraphs** for the same outcome (multiple chemotypes, multiple network signatures, multiple linguistic patterns), and our MoE is designed to reflect this by letting different experts specialize in different causal subgraphs and letting the gate choose, per instance, which causal explanation is most appropriate.
>
> We will edit the manuscript to have uniform term usage for clarity and more elaboration for the instance-level heterogeneity realization in each dataset used.
>
> ...Continued in next post.

---

> ### Author Response · Authors · 2025-11-19
> **Response to Reviewer nffD (Part 2)**
>
> ...Continued from previous post.
>
> **W2\Q2: Assumption 3.5**
>
> > *My understanding of the method is as follows: different experts are used to extract different subgraph structures based on a diversity constraint. A selection mechanism then assigns the maximum predictive weight to the expert with the lowest prediction loss, and the subgraph extracted by this expert is considered the causal part. Is this understanding correct?*
> > *… it implicitly requires that the causal subgraph structures present in the test data must have been seen during training and learned by at least one expert.*
>
> We thank the reviewer for the careful reading and for the insightful question, which helps us further clarify our method and problem setting. The reviewer’s interpretation is essentially correct, with a small clarification:
>
> 1. The gate does **not** literally choose “the expert with the lowest loss” in a hard argmin sense; instead, it is a **learned soft-selection mechanism** whose output probabilities are shaped by (i) the task loss of each expert, and (ii) the sparsity constraint. At convergence, the expert(s) receiving the highest gating weight for an instance tend to be the ones that consistently achieve low prediction loss on that type of instance.
>
> Regarding **Assumption 3.5**, the reviewer is correct in noting that it implies: for each test instance, there exists at least one expert whose learned subgraph structure can explain that instance well. However, we respectfully argue that this is *not* an unusually strong condition in our setting, but rather the natural analogue of standard covariate shift assumptions:
>
> - As stated in Section 2, we explicitly focus on **covariate-type OOD shifts** which, following existing literature (LECI, Gui et. al., 2023; GraphMetro Wu et. al., 2024; GSAT Miao et. al., 2022) and the benchmark that we use (GOOD, Gui et. al., 2022), is defined as follows:
>   -  Covariate shift occurs when the relationship between the input graph and its associated label is stable but the distribution of graph inputs changes.
>   - Formally, this is when $P_{train}(X) \neq P_{test}(X)$ but $P_{train}(Y \vert X) = P_{test}(Y \vert X)$.
>   - Under the causal subgraph view, this means that the mapping $G_c \to Y$ must remain the same across training and testing environments (IRM, Arjovsky et. al. 2019; GOOD, Gui et. al. 2022).
>   - If a new causal subgraph appeared in the test set that never appeared in training, then the model would face a new, unseen mapping $G_c \to Y$ which would violate the above invariance assumption.
>   - Therefore, introducing new causal subgraphs in the test environment would change the semantics of the prediction task, which correspond to semantic/concept drift, not covariate shift.
>
> Returning back to Assumption 3.5:
> - Again, the reviewer is correct is in understanding that this assumption implicitly requires that the causal subgraph structures present in the test environment be seen during training. This requirement follows directly from the nature of the covariate shift setting.
> -  In this sense, Assumption 3.5 is analogous to the usual **support-overlap/coverage assumption** in covariate shift: the *causal part* of test instances must lie within the support of what the model has seen during training.
> - Practically, this is exactly the setup of the GOOD benchmark that we use (Gui et. al., 2023):
>     - In **GOOD-HIV (Scaffold/Size)** and **GOOD-Motif (Basis/Size)**, the OOD shifts are driven by changes in *scaffolds, sizes, or motif distributions* in the **spurious** or surrounding structure, while the underlying causal subgraphs (chemotypes or motif types) remain present in the training environments. For example, in GOOD-Motif each graph contains exactly one of three motifs (house, directed cycle, or crane), and the task is to identify which motif is present. The train–test difference lies in the *spurious graph* around the motif: a “house’’ motif might be attached to a tree-like base graph during training but to a ladder-like base graph at test time. Thus, the causal subgraphs in the test data (house/cycle/crane) are already seen during training; what changes across training and testing environments is the surrounding spurious structure.
>   - In **GOOD-Twitter** and **GOOD-SST2**, the label can be explained by multiple structural signatures (network patterns or syntactic patterns), but those signatures are consistent between the training and testing environments; the OOD shift comes from changing the length distribution of the input, not from introducing a completely new and unseen causal subgraph at test time.
>   - In **CFP-Motif (Covariate/FIIF/PIIF)**, the distribution shifts are generated in a similar manner as GOOD-Motif except with three different causal assumptions which differ in **how spurious features are generated relative to the causal motif and label**; the causal subgraphs themselves are shared across training/testing environments.
>
> ...Continued in next post.

---

> ### Author Response · Authors · 2025-11-19
> **Response to Reviewer nffD (Part 3)**
>
> ...Continued from previous post.
>
> Our MoE architecture is designed precisely for this type of shift (covariate):
>
> - Different experts specialize in different **causal subgraphs**.
> - The gate learns to **select per-instance** which expert’s subgraph is most predictive.
> - Assumption 3.5 simply states that, for each test instance, *the relevant causal subgraph is among those that at least one expert has already learned* — a standard and practically satisfied condition on the GOOD and CFP-Motif benchmarks, where shifts are implemented by changing **spurious structure** rather than by introducing entirely novel causal graphs.
>
> Lastly, if the test environment were allowed to introduce **entirely new causal subgraphs that never appear in training**, then *no* purely observational method (MoE or otherwise) could guarantee OOD generalization without additional supervision or strong extrapolative assumptions. Such a setting corresponds to semantic or concept drift, which is a fundamentally different problem from the covariate shift scenario considered here.
>
> ---
>
> **W3: Analysis of Computational Efficiency**
>
> We thank the reviewer for this suggestion. In the following table, we present a computational efficiency analysis by varying $K \in \{1,4,8,16,32\}$ and measuring training time, inference time, and OOD accuracy on three representative datasets:
>
>
> | Dataset      | $K$ | Training time | Inference time | OOD Acc   |
> |-------------|-----------------:|:-------------:|:--------------:|:-----:|
> | **Twitter** | 1   | 3m12s  | 4s  | 58.68 |
> |             | 4   | 3m18s  | 4s  | 60.48 |
> |             | 8   | 3m30s  | 4s  | **61.13** |
> |             | 16  | 3m47s  | 5s  | 60.05 |
> |             | 32  | 4m34s  | 6s  | 60.12 |
> | **SST2**    | 1   | 38m35s | 26s | 81.11 |
> |             | 4   | 45m19s | 42s | 83.41 |
> |             | 8   | 58m47s | 50s | **83.73** |
> |             | 16  | 1h15m21s | 50s | 83.72 |
> |             | 32  | 1h56m05s | 1m20s | 83.92 |
> | **Motif-basis** | 1 | 13m45s | 6s  | 83.90 |
> |             | 4   | 15m38s | 6s  | 91.79 |
> |             | 8   | 17m07s | 6s  | **92.80** |
> |             | 16  | 22m34s | 10s | 92.47 |
> |             | 32  | 31m16s | 12s | 92.33 |
>
>
>
>
> Notably, both training and inference times increase linearly (and in practice, sub-linearly) with $K$ while OOD accuracy remains largely stable with slight improvements/degradations depending on the dataset.
>
> Our model is intentionally designed to scale modestly with respect to the number of experts:
> - All experts share a **single subgraph extractor** with different predictor heads for subgraph extraction.
> - Each expert itself is lightweight, consisting of a single GNN.
>
> For further discussion of computational complexity and scalability, please see our response to W3 of Reviewer GS29.
>
> ---
>
> We would like to thank the reviewer again for their insightful questions. We will incorporate the clarifications and definitions provided in this response into the manuscript, and we appreciate the reviewer’s help in making the paper clearer and stronger.

---

### Official Review · Reviewer_q7vv · 2025-10-18

**Soundness:** 3
**Presentation:** 3
**Contribution:** 3
**Rating:** 6
**Confidence:** 4

**Summary:**

This work introduces a Mixture-of-Experts (MoE) approach for addressing out-of-distribution graph classification tasks. Intuitively, each expert would learn a (perhaps disjoint) subset of the environment distributions. By softly enforcing diversity and sparseness of the expert distribution, the authors ensure that the resulting method both covers the environment space and selects the correct expert for that environment with high probability. Experimental results highlight the effectiveness of the method for graph classification.

In summary, the work is nicely written, the theoretical analysis (although simple) provides a clear understanding of the method, empirical results are promising, and notation is clean. Although the manuscript presents minor issues (e.g., $l_{sparse}$ is only described in the appendix - with a different name), I am happy to provide a favorable review for the work.

**Strengths:**

1. The method (although unnamed) is clearly described.

2. The theoretical analysis is to-the-point and provides an intuitive understanding for the chosen design.

3. Empirical results highlight the method’s competitive performance.

4. Ad-hoc terminology (e.g., semantic diversity, mechanism coverage) is rigorously defined.

**Weaknesses:**

1. I believe that the main disadvantage of the introduced design is the introduction of many hyperparameters for model regularization. In practice, how can they be selected? Although there are standard approaches in machine learning for hyperparameter selection, these are often expensive.

2. Also, I am a bit doubtful about the use of the word OOD; usually, this means a difference between training and test datasets. It is unclear how it is represented in the experimental analysis in this section. Could the authors elaborate briefly on this?

3. Diversity and sparsity constraints seem to solely lead to marginal improvements over the baseline method - which also presents competitive performance regardless. Are there datasets for which the imposed constraints lead to more significant performance boosts?

4. Could the authors discuss the runtimes for the presented methods in Table 1? Also, when adding further constraints (e.g., diversity and sparsity), the optimization problem changes; does it significantly affect the convergence of the stochastic gradient algorithm?

5. [Minor] $l_{sparse}$ in line 304 uses a different notation in the appendix.

6. [Minor] I could not find the anonymous repository referred to in the manuscript. Could the authors share the code?

**Questions:**

Please refer to the questions above.

---

> ### Author Response · Authors · 2025-11-19
> **Response to Reviewer q7vv (Part 1)**
>
> We are grateful to the reviewer for their time and thoughtful feedback. Our detailed responses are provided below.
>
> ---
>
>
> **W1 Hyperparameter Selection**
>
> **Number of hyperparameters.**
>
> We agree that hyperparameter selection is an important practical concern in deep learning. However, we do not believe the *number* of hyperparameters in our method is excessive or a particular weakness:
>
> - Our method uses **no more hyperparameters (and in some cases fewer)** than other SOTA graph OOD methods. For example, LECI must tune the losses of two adversarial networks and two predictor networks, as well as two adversarial training schedules (Gui et. al., 2023). Similarly, the MoE-based GraphMetro needs to choose which augmentations to apply (node add/drop, edge add/drop, subgraph sampling, feature noise, etc.), their magnitudes, and the number of experts to use (Wu et. al., 2024).
> - Other than the coefficients that balance the loss terms, an approach used by many prior methods (e.g., LECI and GraphMETRO), our model introduces only two method-specific hyperparameters: the edge keep-rate and the number of experts $K$.
> - Empirically, our model is **not highly sensitive** to these method-specific hyperparameters (edge keep rate and $K$); performance remains stable under reasonable variations. This is supported by the sensitivity analyses reported in RQ4 of the manuscript.
>
> **Hyperparameter tuning.**
>
> Regarding how we select hyperparameters in practice:
>
> - For the loss terms, we did **not** perform extensive balancing; we simply ensured that the task loss (cross-entropy) is not dominated by the regularization terms. More advanced gradient-balancing techniques could be used if desired, as discussed in our response to W2 for Reviewer GS29, but we found them unnecessary in our experiments. This practice is consistent with many deep learning methods, including the graph OOD baselines we compare against, which also combine multiple losses during training.
> - For the main training hyperparameters, we **follow the GOOD benchmark protocol** and only tune **three** hyperparameters—batch size, learning rate, and edge keep rate—using a **small budget of 10 trials**. All other architectural hyperparameters (hidden dimension, model depth, etc.) are kept fixed and aligned with the GOOD benchmark baselines (Gui et. al., 2022).
>
> Taken together, these points show that while our method has the standard deep-learning hyperparameters, **it does not require heavy or costly tuning**. The overall hyperparameter set is comparable to prior work, the method is stable under reasonable changes, and all reported results were obtained with a modest tuning budget (following GOOD benchmark protocols).
>
> ---
>
> **W2: GOOD as an OOD dataset**
>
> **Usage of a standard OOD benchmark.**
> In all our experiments, *OOD* refers specifically to a **systematic distribution shift between training and test data**. This shift is instantiated through the GOOD benchmark (Gui et. al., 2022), which has become a standard testbed for graph OOD and graph generalization. In particular, GOOD is also adopted by several recent graph OOD methods such as CIGA (Chen et. al., 2022), GSAT (Miao et. al., 2022), LECI (Gui et. al., 2023), and Graph Structure Extrapolation (GSE; Li et. al., 2024), all of which evaluate primarily on GOOD-HIV, GOOD-Motif, GOOD-SST2, and related splits. Our use of GOOD and its environment-based splits therefore follows the prevailing experimental protocol in the graph OOD literature, rather than introducing a custom or non-standard OOD setup.
>
> ...Continued in next post.

---

> ### Author Response · Authors · 2025-11-19
> **Response to Reviewer q7vv (Part 2)**
>
> ...Continued from previous post.
>
> **How OOD shift is realized in GOOD.**
> For each dataset, GOOD constructs **source environments** (used for training/validation) and **target environments** (used only for testing) that differ along a particular structural or semantic factor. In this paper we use four GOOD datasets with covariate shifts (described in detail in Appendix E), each with a well-defined type of shift between train and test:
>
> - **GOOD-HIV (Scaffold / Size).**    Each graph is a molecule; the task is to predict HIV inhibition.
>   - *HIV-Scaffold:* train and test environments differ in **core molecular scaffolds**,
>   - *HIV-Size:* train and test environments differ in the **distribution of molecule sizes**.
>
> - **GOOD-Motif (Basis / Size).**    Synthetic graphs are built by attaching motifs to random base graphs; labels depend on the motif.
>   - *Motif-Basis:* train and test environments use **different sets of base graphs**,
>   - *Motif-Size:* train and test environments differ in base graph **size/density**.
>
> - **GOOD-Twitter.**    Each graph is a user ego-network; the task is to classify a user attribute. Train and test environments correspond to different **user communities**, which mainly differ in length.
>
> - **GOOD-SST2.**    Each sentence is represented as a dependency parse tree; the task is sentiment classification. Train and test environments differ in the distribution of **sentence lengths.**
>
> Across these four tasks, the OOD setting always arises from **deliberate shifts between training and test environments**, rather than from random splits of i.i.d. data. Our reported “OOD accuracy’’ is always measured on these environments with unseen shifts.
>
> For more details regarding the dataset generation of these datasets, please refer to the original GOOD benchmark paper (Gui et. al., 2022).
>
> ---
>
> **W3: Importance of Diversity and Sparsity**
>
> We thank the reviewer for raising an important point regarding the roles of diversity and sparsity in our method. Diversity ($L_{\mathrm{div}}$) and sparsity/gating ($L_{\mathrm{gate}}$) are core components of our MoE approach. To further clarify their impact, we provide additional ablations (extending Table 3 in the paper to more datasets and more ablated loss terms), which include several cases where removing a component leads to a substantial reduction in performance.
>
>
> | Dropping                  | Twitter | SST2  | Motif Basis | Motif Size | HIV Scaffold | HIV Size |
> |---------------------------|:-------:|:-----:|:-----------:|:----------:|:------------:|:--------:|
> | $L_{\mathrm{div}}$        | 60.10   | 82.20 | 91.13       | 70.07      | 65.95        | 65.23    |
> | $L_{\mathrm{gate}}$       | 59.92   | 81.97 | 89.60       | 74.36      | 68.56        | 61.91    |
> | $L_{\mathrm{reg}}$        | 60.60   | 83.46 | 67.48       | 73.75      | 68.55        | 64.79    |
> | **Standard (full loss)**  | **61.13** | **83.73** | **92.80**     | **75.52**    | **71.55**      | **66.98**  |
>
> For example:
>
> - Removing $L_{\mathrm{div}}$ on **HIV-Scaffold** reduces accuracy from 71.55 → 65.95.
> - Removing $L_{\mathrm{gate}}$ on **HIV-Size** reduces accuracy from 66.98 → 61.91.
>
> While some datasets show a smaller degradation, the overall pattern is clear: each loss component plays a role in achieving strong performance, and the full model consistently performs best.
>
>
> **Impact on ranking.**
> Even when the absolute drop is “only’’ a few points, it is far from negligible on the GOOD benchmark. For example:
>
> - On **HIV-Size**, going from 66.98 (full model) to 64.79 (no $L_{\mathrm{reg}}$) moves our method from **1st to 5th** in Table 1.
> - On **HIV-Scaffold**, going from 71.55 (full model) to 68.55 (no $L_{\mathrm{reg}}$) moves us from **4th to 11th** in Table 1.
>
> ---
> **W4: Runtime and Convergence**
>
> We thank the reviewer for bringing up a valuable point regarding the runtime and convergence of our training algorithm.
>
> **Runtime.**
>
> The exact runtimes in Table 1 are not directly comparable, as they depend heavily on implementation details and hyperparameters such as batch size. Nevertheless, our method exhibits favorable computational characteristics, both in terms of having a comparable hyperparameter count relative to SOTA methods like LECI and in terms of computational efficiency scaling with respect to the number of experts, as discussed in our response to Reviewer GS29 (W4).
>
> ...Continued in next post.

---

> ### Author Response · Authors · 2025-11-19
> **Response to Reviewer q7vv (Part 3)**
>
> ...Continued from previous post.
>
> **Convergence.**
>
> We did not observe any training instability or divergence in our experiments. We provide [training statistics on the Motif-basis dataset (click here)](https://ibb.co/gbqQ8t5H). *Note that we employ uniform gating for the first ten epochs for training stability so we show diversity/gate loss starting at epoch 10 (when we begin encouraging specialization).*
>
> We attribute this stability to two design choices:
>
> 1. **Uniform gating during the first $k$ epochs**: This ensures that all experts receive sufficient training signal before the gating network begins selective routing. This warm-up phase ensures that each expert learns a reasonable initial representation, which stabilizes early optimization.
>
> 2. **Module-specific loss assignment to reduce gradient interference:** We apply different loss terms to different modules, which reduces potential gradient conflicts:
>    - The **gate** only receives the gating loss.
>    - The **experts** only receive the task loss (cross-entropy).
>    - The **subgraph extractor** is the only component that receives multiple objectives (task, diversity, and regularization losses).
>
> For additional discussion on balancing the contribution of each loss term, please see our response to W2 of Reviewer GS29.
>
> ---
>
> **W5 and W6 Typo and Broken Repository Link**:
> Thank you for catching the typo. The hyperlink should be clickable, but we also include the URL here: https://anonymous.4open.science/r/MoEOOD-B60A/
>
>
> ---
>
> We appreciate the reviewer’s careful reading and thoughtful comments. We will integrate the additional results and clarifications into the revised manuscript, and we are grateful for the suggestions that help improve the paper.

---

### Official Review · Reviewer_GS29 · 2025-10-25

**Soundness:** 3
**Presentation:** 2
**Contribution:** 2
**Rating:** 4
**Confidence:** 3

**Summary:**

The paper introduces a novel MoE framework designed to extract diverse subgraphs and address dataset heterogeneity. The proposed approach theoretically demonstrates that both semantic diversity among experts and sparse expert selection contribute to reducing prediction error. Extensive experiments are conducted to validate the framework’s effectiveness across different datasets, highlighting its potential for improving generalization in heterogeneous graph learning scenarios.

**Strengths:**

1. The paper proposes a novel MOE framework for extracting diverse subgraphs, aiming to address dataset heterogeneity.

2. A theoretical analysis is provided to demonstrate that semantic diversity among experts, together with sparsity in expert selection, jointly contributes to error reduction.

3. Extensive experiments are conducted to validate the effectiveness of the proposed approach.

**Weaknesses:**

1. It is necessary to clarify which components are genuinely novel, and whether the proposed method is merely a combination of existing MoE techniques.

2. The total loss comprises multiple terms, so it would be helpful to explain how their relative influences are balanced. More detailed ablation studies are needed to further validate the design, and to examine whether each component is essential for the final performance.

3. Considering the trade-off between performance and computational cost, how many experts are recommended to be used? A detailed analysis or estimation of computational efficiency would also strengthen the work.

**Questions:**

None, see weaknesses.

---

> ### Author Response · Authors · 2025-11-19
> **Response to Reviewer GS29 (Part 1)**
>
> We thank the reviewer for their time and insightful comments. We address each point in detail below.
>
> ---
>
> **W1: Genuine novelty**
>
> We thank the reviewer for the opportunity to highlight the novelties of our method:
>
> - As Reviewer 4z1L highlighted in their Strength 2, this is the first MoE method that tackles graph OOD via **causal subgraphs** [Note 1]. Since our method is the first MoE architecture explicitly designed to model diversity at the *causal subgraph* level, it can handle heterogeneous causal subgraphs (HCS) by training different experts to specialize on different causal subgraphs **without** using environment labels or any other spurious subgraph metadata.
> - We further provide a **novel theoretical framework** that links OOD risk to **semantic diversity** and **instance-level sparsity** among experts. In particular, Theorem 3.6 shows how sparsity and diversity contribute to OOD risk through explicit *selection* and *coverage* terms.
>
> - Our method is robust to **heterogeneous causal assumptions (HCA)** (covariate, FIIF, PIIF). This is achieved because we do **not** impose restrictive assumptions on the underlying structural causal model (SCM)—i.e., we do **not** assume a specific way in which the spurius subgraph relates to the class label, the environment label, or the causal subgraph. This “assumption-light’’ design is what allows our method to perform strongly across different causal assumption settings; see the new CFP-Motif results in our response to W2 (Reviewer 4z1L).
>
>
> - Finally, these contributions are reflected empirically: our method achieves **superior OOD performance across a broad range of datasets**, consistently outperforming prior graph OOD baselines.
>
> **Notes.**
>
> [1] **MoE for OOD.**  The only other MoE-based approach for graph OOD that we are aware of, GraphMETRO (Wu et. al., 2024), uses various graph transformations to create multiple augmented views of each graph. This design is not a causal subgraph approach; instead, it is a data augmentation approach, which can be susceptible to label-validity issues introduced by the augmentations.
>
> [2] **HCS vs. HCA.**  *Heterogeneous causal subgraphs (HCS)* refers to the phenomenon where **causal subgraphs** varies across individual samples. *Heterogeneous causal assumptions (HCA)* refers to the underlying **structural causal model (SCM)** of the dataset. The SCM for a dataset specifies how the causal subgraph, the spurious subgraph, the environment label, and the final task label are generated and related to one another.
>
> ---
>
> **W2: Multiple Loss Terms**
>
> **Balancing loss terms.**
>
> We thank the reviewer for raising the question about how the different loss terms are balanced. Using multiple loss components is standard practice in deep learning and, in itself, should not be viewed as a weakness. That said, in our case, the way these losses are applied and balanced is structurally simple (visually illustrated in Figure 1 of our paper):
>
> - The **gate** only receives the gating loss.
> - The **experts** only receive the task loss (cross-entropy).
> - The **subgraph extractor** is the only component that receives multiple objectives (task, diversity, and regularization losses).
>
> Thus, most parts of the model are optimized with a *single* loss term, and only the subgraph extractor is trained under a multi-objective signal. In practice, we combine these losses using fixed scalar weights (a common baseline in multi-task learning) and did not observe instability or sensitivity issues in our experiments. For more discussion on convergence and stability, see our response to W4 Reviewer q7vv.
>
> If needed, our approach can be extended with established gradient-balancing techniques from multi-task learning—such as uncertainty-based loss weighting (Kendall et al., 2018), gradient norm balancing (Chen et al., 2018), or conflict-aware “gradient surgery’’ methods that modify gradients based on their pairwise angles (e.g., PCGrad; Yu et al., 2020)—to further mitigate potential conflicts between loss gradients. In practice, however, **our method already exhibits stable training dynamics and good convergence across all experiments**, so we did not find it necessary to employ any of these additional mechanisms.
>
> ...Continued in next post.

---

> ### Author Response · Authors · 2025-11-19
> **Response to Reviewer GS29 (Part 2)**
>
> ...Continued from previous post.
>
> **Importance of multiple loss terms.**
>
> We thank the reviewer for suggesting a more detailed analysis of the loss components; this motivated us to perform additional ablations that further highlight the benefit of each loss term. Our overall objective combines four terms: $L_{\mathrm{ce}}$, $L_{\mathrm{reg}}$, $L_{\mathrm{div}}$, and $L_{\mathrm{gate}}$ [Note 1].
>
> To examine whether each component is essential, we remove **one** loss term at a time while keeping all other settings fixed. The table below reports the resulting OOD accuracies:
>
> | Dropping              | Twitter | SST2  | Motif-basis | Motif-size | HIV-scaffold | HIV-size |
> |-----------------------|:-------:|:-----:|:-----------:|:----------:|:------------:|:--------:|
> | $L_{\mathrm{div}}$    | 60.10   | 82.20 | 91.13       | 70.07      | 65.95        | 65.23    |
> | $L_{\mathrm{gate}}$   | 59.92   | 81.97 | 89.60       | 74.36      | 68.56        | 61.91    |
> | $L_{\mathrm{reg}}$    | 60.60   | 83.46 | 67.48       | 73.75      | 68.55        | 64.79    |
> | **Standard (full loss)** | **61.13** | **83.73** | **92.80**     | **75.52**    | **71.55**      | **66.98**  |
>
> Across all datasets, removing *any* of the loss components results in a performance drop relative to the full model. In several cases, the decrease is very significant—for example:
>
> - Dropping $L_{\mathrm{reg}}$ on Motif-basis (from 92.80 → 67.48).
> - Dropping $L_{\mathrm{div}}$ on HIV-scaffold (71.55 → 65.95).
> - Dropping $L_{\mathrm{gate}}$ on HIV Size (66.98 → 61.91).
>
> While for some datasets the degradation is less significant, the overall trend is consistent: *each loss component contributes to achieving strong performance, and the full model performs best across the board.* These ablations support our design choice: each loss term contributes meaningfully to the final OOD performance.
>
> **Notes.**
>
> [1] **Loss Terms.**
> - $L_{\mathrm{ce}}$: standard task loss (cross-entropy).
> - $L_{\mathrm{reg}}$: regularizes the size of the extracted causal subgraph $G_c$ (penalizing degenerate cases where $G_c = G$ or $G_c$ is empty).
> - $L_{\mathrm{div}}$: encourages diversity among the extracted $G_c$’s across experts.
> - $L_{\mathrm{gate}}$: promotes sparse and selective expert routing.
>
>
> ---
> **W3: Number of Experts ($K$), Performance, and Computational Trade-off**
>
> Overall, $K = 8$ provides a good balance between accuracy and cost.
>
> To further show accuracy and computational trade-off, we vary $K \in \{1,4,8,16,32\}$ and measure training time, inference time, and OOD accuracy on three representative datasets:
>
> | Dataset      | $K$ | Training time | Inference time | OOD Acc   |
> |-------------|-----------------:|:-------------:|:--------------:|:-----:|
> | **Twitter** | 1   | 3m12s  | 4s  | 58.68 |
> |             | 4   | 3m18s  | 4s  | 60.48 |
> |             | 8   | 3m30s  | 4s  | **61.13** |
> |             | 16  | 3m47s  | 5s  | 60.05 |
> |             | 32  | 4m34s  | 6s  | 60.12 |
> | **SST2**    | 1   | 38m35s | 26s | 81.11 |
> |             | 4   | 45m19s | 42s | 83.41 |
> |             | 8   | 58m47s | 50s | **83.73** |
> |             | 16  | 1h15m21s | 50s | 83.72 |
> |             | 32  | 1h56m05s | 1m20s | 83.92 |
> | **Motif-basis** | 1 | 13m45s | 6s  | 83.90 |
> |             | 4   | 15m38s | 6s  | 91.79 |
> |             | 8   | 17m07s | 6s  | **92.80** |
> |             | 16  | 22m34s | 10s | 92.47 |
> |             | 32  | 31m16s | 12s | 92.33 |
>
> **Computational efficiency.**
> Our MoE design keeps the cost growth modest:
>
> - All experts share a **single subgraph extractor**; only the predictor head is replicated per expert.
> - Expert computations can be **parallelized** during both training and inference.
>
> Consequently, both training and inference times increase linearly (and in practice, sub-linearly) with $K$, as seen in the table above. This empirical behavior matches the intended design: using more experts does **not** lead to computational explosion.
>
>
> **OOD Accuracy.**
>
> Further increasing $K$ beyond 8 experts yields only marginal gains (SST2, Motif-basis) or even slight degradation (Twitter). The drop on Twitter is likely due to scaling-law effects (i.e., the capacity–data trade-off): Twitter is much smaller (≈3k samples vs. ≈70k and ≈30k for SST2 and Motif-basis), so using a very large number of experts can over-parameterize the model relative to the data and slightly hurt generalization.
>
> ---
>
> We thank the reviewer for their constructive feedback. We will include the additional results and discussion in the updated version of the paper, and we hope the clarifications provided here address the reviewer’s concerns.

---

> > ### Comment · Reviewer_GS29 · 2025-11-20
> >
> > I thank the authors for their further clarification. The newly added experimental results convincingly demonstrate the effectiveness of their method.

---

### Official Review · Reviewer_4z1L · 2025-11-01

**Soundness:** 4
**Presentation:** 4
**Contribution:** 4
**Rating:** 6
**Confidence:** 4

**Summary:**

The paper proposes a causal subgraph–based MoE framework for graph OOD generalization. The key idea is to consider instance-level different causal subgraph conceptual experts. The authors give a risk decomposition showing that diversity ensures at least one expert matches the unseen environment, and sparsity ensures the gate actually picks it. Empirically, on the GOOD benchmark, the method achieves the best average rank and clearly improves on strong OOD baselines such as LECI and GALA.

**Strengths:**

1. The paper correctly identifies that many causal-subgraph OOD methods assume one invariant causal subgraph per task/environment, but different instances often rely on different motifs.
2. The idea of learning different causal subgraphs using MoE is fresh. I will explain from a different perspective, though: different MoE learns different spurious correlations. Given enough experts, there will always be one expert learning no spurious correlations with the diversity constraint.
3. The experimental results are very strong with useful and complete ablations.
4. The assumption is lighter, easier to extend.

**Weaknesses:**

1. Given the story, there is few interpretability results in terms of how MoE solves the instance-level heterogeneity problem.
2. LECI provides several enhanced Motif datasets, CFP-Motif, which represent different causal assumptions. The authors may train the three splits together. If the MoE assumption stands, we should observe that, given different underlying causal assumptions, the gate activation should follow clear patterns. Without this experiment or other alternative, the claim will not be justified enough.
3. The authors show 1, 4, 8 experts, but in MoE literature there is always a question: does diversity collapse or does routing become unstable when K=16/32 on small datasets?
4. Lack of visualizations of learned subgraphs per expert.

**Questions:**

N/A

---

> ### Author Response · Authors · 2025-11-19
> **Response to Reviewer 4z1L (Part 1)**
>
> We thank the reviewer for their time and effort in providing valuable feedback on our work. Below, we provide our responses.
>
> ---
>
> **W1: Lack of interpretability for instance-level heterogeneity**
>
> We thank the reviewer for this comment, which prompted us to clarify and empirically demonstrate how our MoE framework handles instance-level heterogeneous causal subgraphs (HCS). This response also partially addresses the reviewer’s concern in W2.
>
> **Purpose.**
>
> - **P1.** Show that the gate utilizes some (but not all) experts for each causal subgraph.
> - **P2.** Show that the gate utilizes different experts for different causal subgraphs.
> - **P3.** Show that the gate assign experts to the subgraph that they specialize in.
>
> **Setup.**
>
> - We train our MoE model on the Motif-basis dataset, which has three heterogeneous causal subgraphs (i.e., three different motifs).
> - On the test set, we compute the average per-expert, per-causal-subgraph accuracy and usage (i.e., gate probability) over all instances that contain the corresponding causal subgraph. This yields $m \times k$ (accuracy, usage) pairs; in this experiment, $m = 3$ causal subgraphs and $k = 8$ experts.
>
>
> **Results.**
>
>
> For an interpretable visualization of our results, we present:
>
> **[Three bar plots showing per-expert usage across three heterogeneous causal subgraphs (click here)](https://ibb.co/zVqPyJnR)**
>
> This bar plot addresses **P1** and **P2**.
> For **P1**, for causal subgraph 1 (bar plot 1), experts 4, 6, and 7 are hardly used.
> For **P2**, focusing on expert 6, although it is hardly used for subgraphs 1 and 3, it is one of the most used experts for subgraph 2.
>
> Overall: Causal subgraph 1 mainly utilizes experts 1/2/3/8. Causal subgraph 2 mainly utilizes experts 1/4/5/6/7. Causal subgraph 3 mainly utilizes experts 3/4/5/7/8.
>
> **[A scatter plot of usage vs. accuracy across all experts and causal subgraphs (click here)](https://ibb.co/dwWd9sxz).**
>
>
> For **P3**, The scatter plot shows that the gating network consistently directs instances toward experts that exhibit strong accuracy on classifying the corresponding causal subgraph. This is evident from the strong positive correlation between usage and accuracy: a Pearson correlation test yields $r = 0.823$ with $p < 10^{-7}$.
>
>
>
> ---
>
> **W2: Test on enhanced motif datasets from LECI (CFP-Motif)**
>
> We thank the reviewer for this suggestion. This additional experiment helps clarify that our method can handle **heterogeneous causal assumptions (HCA)** (covariate, FIIF, PIIF), in addition to the **heterogeneous causal subgraphs (HCS)** we already analyze in our response to W1. It is important to distinguish these two notions—HCS vs. HCA; we elaborate on this in Note [1]. This comment focuses on HCA, while HCS is addressed in our response to W1 (Reviewer 4z1L).
>
> **Setup.**
>
> We evaluate our method on the **CFP-Motif** benchmark introduced by LECI (Gui et. al., 2023), which consists of three datasets, each corresponding to a different causal assumption: Covariate, FIIF, and PIIF (see Note [2] for details).
>
> For all three variants, we **train and evaluate independently** using the **same model and training protocol**, with **$K = 8$ experts**.
>
> ...Continued in next post.

---

> ### Author Response · Authors · 2025-11-19
> **Response to Reviewer 4z1L (Part 2)**
>
> ...Continued from previous post.
>
> **Results.**
>
> The table below reports OOD accuracy (mean $\pm$ standard deviation) on CFP-Motif. Baseline numbers are taken from LECI (Gui et. al., 2023). Our results are reported over five runs.
>
> | Method          | Covariate       | FIIF            | PIIF            |
> |----------------:|:----------------|:----------------|:----------------|
> | ERM             | 57.56 (9.59)    | 37.22 (3.70)    | 62.45 (9.21)    |
> | IRM             | 58.11 (5.14)    | 44.33 (1.52)    | 68.34 (10.40)   |
> | VREx            | 48.78 (7.81)    | 34.78 (1.34)    | 63.33 (6.55)    |
> | CORAL           | 57.11 (8.35)    | 42.68 (7.09)    | 60.33 (8.85)    |
> | DANN            | 49.45 (8.05)    | 43.22 (6.64)    | 62.56 (10.39)   |
> | DIR             | 44.67 (0.00)    | 42.00 (6.77)    | 47.22 (8.79)    |
> | GSAT            | 68.22 (7.23)    | 51.56 (6.59)    | 61.22 (8.80)    |
> | CIGA            | 56.78 (2.99)    | 39.11 (7.70)    | 45.67 (7.52)    |
> | LECI            | 83.20 (5.89)    | 77.73 (3.85)    | 69.40 (7.54)    |
> | **Ours (K = 8)**| **90.83 (1.73)**       | **84.17 (3.88)**       | **77.19 (6.04)**       |
>
> Our method outperforms LECI and all other baselines across **all three** causal assumptions. Here, unlike in our response to W1, we do **not** claim that the MoE architecture alone is responsible for the gains nor do we claim that different experts specialize in different causal assumptions. Rather, we attribute a key part of the improvement to the fact that our approach **does not rely on auxiliary invariance losses tied to a specific causal assumption** (e.g., enforcing invariant $G_s$ across labels, as in LECI (Gui et. al., 2023)). This is what we mean by our method being **assumption-light**: it does not encode whether the underlying structural causal model (SCM) follows covariate shift, FIIF, or PIIF, neither in the loss functions nor in the model architecture.
>
> **Notes.**
>
> [1] **HCS vs. HCA.**
> - *Heterogeneous causal subgraphs (HCS)* refers to the phenomenon where **causal subgraphs** varies across individual samples. *Heterogeneous causal assumptions (HCA)* refers to the underlying **structural causal model (SCM)** of the dataset. The SCM for a dataset specifies how the causal subgraph, the spurious subgraph, the environment label, and the final task label are generated and related to one another.
>
> [2] **Different Causal Assumptions in CFP-Motif.**
> - **Covariate:** classic covariate shift, where the spurious structure changes across environments but is conditionally independent of the label given the causal subgraph.
> - **FIIF:** “fully informative invariant features,” where the spurious part depends on the causal subgraph but does not carry additional information about label.
> - **PIIF:** “partially informative invariant features,” the spurious subgraph is partially explained by the causal subgraph but also has independent correlation with the label.
>
> ...Continued in next post.

---

> ### Author Response · Authors · 2025-11-19
> **Response to Reviewer 4z1L (Part 3)**
>
> ...Continued from previous post.
>
> ---
>
> **W3: Diversity Collapsing & Routing Instability for $K = 16/32$**
>
> Both expert diversity and routing stability remain consistent when increasing the number of experts.
>
> **Diversity.**
>
> Semantic diversity (i.e., the ability of different experts to extract different candidate causal subgraphs) is enforced by encouraging different experts to extract de-correlated masks. On the Motif-basis dataset, [we show that the diversity loss for both $K = 8$ and $K = 32$ experts steadily decreases during training (click here)](https://ibb.co/21K0Ckdz). *Note that we employ uniform gating for the first ten epochs for training stability so we show diversity loss starting at epoch 10 (when we begin encouraging specialization).* These results show that diversity is preserved at higher numbers of experts.
>
>
> Moreover, in term of accuracy, specialization of experts (i.e., high loss gap) on different causal subgraphs is maintained. On the Motif-basis dataset, this [figure reports per-expert accuracy on each of the 3 causal subgraphs across 32 experts (click here)](https://ibb.co/bMhXJPzx). The maintained diversity is evident from experts having sharply different accuracy profiles across subgraphs. For example, Expert 29 attains almost $100\%$ accuracy on causal subgraph 2 but nearly zero accuracy on the other two subgraphs. Similarly, Expert 21 achieves almost $100\%$ accuracy on causal subgraph 1 while performing poorly on the remaining subgraphs. This pattern is consistent with meaningful expert specialization rather than expert collapse.
>
> **Routing.**
>
> Routing stability is also maintained when using a larger number of experts.
>
> **Setup.**
> Inspired by Monte Carlo dropout (Gal and Ghahramani, 2015), we measure how much the gating distribution over experts varies for the *same* datapoint on the Motif-basis dataset. Focusing on the last two epochs (where dropout is still active but parameter updates are small), we compute for each sample:
>
> - the Jensen–Shannon (JS) divergence (Lin, 2002) between its gating distribution at the last epoch and the second last epoch, and its dimension-normalized variant NJS;
> - the $\ell_1$ distance (sum of absolute differences) between the two distributions, and its dimension-normalized variant.
>
> and then average these quantities over all datapoints.
>
> The results for $K = 8$ and $K = 32$ experts are:
>
> | Metric                                 | 8 experts  | 32 experts |
> |----------------------------------------|-----------:|-----------:|
> | Per-sample JS (mean)                   | 0.1466     | 0.2830     |
> | Per-sample NJS (mean)                  | 0.07049    | 0.0816     |
> | Per-sample $\ell_1$ sum (mean)         | 0.1136     | 0.0352     |
> | Per-sample normalized $\ell_1$ (mean)  | 0.0142     | 0.0011     |
>
> These values remain small even with $K = 32$ experts, indicating that the gating network’s routing decisions remain stable rather than becoming erratic as the number of experts increases. In particular, JS values in the range $0.15$–$0.3$ are typically viewed as reflecting only mild to moderate distributional shifts (Lin, 2002). Both the $K = 8$ and $K = 32$ models fall within these ranges, confirming that the gate distribution is stable under Monte Carlo dropout.
>
> ---
>
> **W4: Visualization of learnt subgraphs**
>
> We thank the reviewer for the suggestion. Although we previously included some visualizations in the Appendix, we have now expanded them. We provide [illustrations of the subgraphs extracted by a four-expert MoE on three training-set graphs from Motif-basis containing the house motif (click here)](https://ibb.co/Cp1DynzV).
>
> These visualizations clearly show that different experts extract distinct subgraphs, demonstrating semantic diversity. Moreover, for each instance, only one or two experts correctly recover the house motif, highlighting expert specialization. Together, these results strengthen our interpretability evidence visually and further support our claims.
>
> ---
>
> We thank the reviewer for their time and thoughtful feedback. We will incorporate the additional results, discussions, and visualizations provided here into the main paper, and we appreciate the reviewer’s suggestions in helping strengthen our work.

---

### Author Response · Authors · 2025-12-04
**General Response**

We sincerely thank the reviewers and the Area Chair for their time, thoughtful questions, and constructive feedback. Their comments have greatly strengthened the clarity, rigor, and overall quality of our work.

In response, we have made the following revisions to the manuscript:

1. **Additional Dataset: CFP-Motif.**
   We added results on the CFP-Motif datasets [1] to Section 4, showing that our method consistently outperforms baselines under three distinct causal assumptions (covariate, FIIF, PIIF). These results highlight the benefits of our assumption-light design by its demonstrating robustness to different causal assumptions.

2. **Expanded Ablations.**
   Section 4 now includes a more comprehensive set of ablation studies spanning additional datasets. Across all settings, the full model achieves the best results, and removing any loss component can lead to significant performance degradation. This provides clear evidence of the necessity of each component.

3. **Theoretical Computational Efficiency Analysis.**
   Section 3 now includes a theoretical analysis of computational complexity on a forward pass of the entire model, showing that the proposed MoE framework exhibits linear scaling in inference time with respect to the number of experts, demonstrating the scalability of our method.

4. **Clarified Definitions.**
   Sections 1–2 now contain more precise definitions of causal diversity, instance heterogeneity, and causal assumptions, along with concrete examples aligned with the datasets used in our experiments for further clarity. We also standardized terminology usage throughout the paper for consistency.

5. **Typographical Corrections.**
   We corrected minor typographical issues and improved wording for clarity and readability.

---

*Additional Revisions: In the next manuscript update (should the paper be accepted), we plan to incorporate the following revisions from our responses to the reviewers:*

1. **Interpretability Results.**
	We will incorporate the interpretability analyses showing the diverse specialization patterns exhibited by individual experts across different causal subgraphs (see response to *W1 - Reviewer 4z1L*). We will also include the visualizations of the extracted causal subgraphs for each expert (see response to *W4 - Reviewer 4z1L*).

2. **Empirical Computational Efficiency Analysis.**
   Complementing the newly added Theoretical Computational Efficiency Analysis in Section 3, we will also add empirical training and inference time analyses showing sublinear scaling behavior with the number of experts, as presented in our response to *Reviewer GS29 W3.*

3. **Training Stability.**
   We will incorporate training-curve analyses illustrating the stability of each loss component, as shown in our response to *Reviewer q7vv W4*.

We once again thank the reviewers and the Area Chair for the dedication of their time and effort towards the improvement of our work.

---

[1] Gui, S., Liu, M., Li, X., Luo, Y., & Ji, S. (2023). Joint learning of label and environment causal independence for graph out-of-distribution generalization. In Proceedings of the 37th Conference on Neural Information Processing Systems

---

### Meta-Review · Area_Chair_7M44 · 2025-12-16

**Summary:**

This paper tackles an important problem in OOD generalization by addressing instance-level heterogeneity induced by diverse causal subgraphs. The proposed Mixture-of-Experts framework is well motivated and avoids restrictive causal assumptions or potentially harmful data augmentation strategies. By enforcing semantic diversity among experts and using a sparse, adaptive gating mechanism, the method provides a principled way to reduce OOD error, which is further supported by theoretical analysis. Experimental results on the GOOD benchmark demonstrate strong performance across both synthetic and real-world structural shifts. Overall, this work offers a meaningful and practical contribution to OOD generalization and merits acceptance.

**Reviewer Concerns:**

Reviewer 4z1L raised concerns regarding the interpretability of the MoE framework in addressing instance-level heterogeneity, the lack of evidence on gate behavior under different causal assumptions, the potential risks of expert diversity collapse and routing instability with larger numbers of experts, and the absence of visualizations of expert-specific subgraphs. In response, the authors substantially strengthened the rebuttal by adding interpretability analyses that explicitly illustrate instance-level heterogeneity, conducting additional experiments on enhanced motif datasets from LECI (CFP-Motif) to examine gate activation patterns under different causal assumptions, providing a detailed discussion on diversity collapse and routing stability, and including visualizations of learned subgraphs for individual experts. These additions directly address the reviewer’s concerns.

----

Reviewer GS29 raised concerns regarding the novelty of the proposed method relative to existing MoE approaches, the role and necessity of multiple loss terms, and the trade-off between performance and computational cost. In the rebuttal, the authors clarified the key conceptual differences from prior MoE-based methods, provided explanations for the design and balancing of the multiple loss terms, and included a detailed analysis of computational efficiency and expert scalability.


-----

Reviewer q7vv raised concerns regarding the introduction of multiple regularization hyperparameters and their practical selection, the use and interpretation of the term OOD in the experimental setting, the marginal gains brought by diversity and sparsity constraints, as well as runtime and convergence behavior under additional optimization constraints. In the rebuttal, the authors provided detailed explanations on hyperparameter selection strategies, clarified the use of a standard OOD benchmark, presented additional experimental evidence on the importance of diversity and sparsity, and discussed runtime and convergence properties, while also addressing minor issues such as notation inconsistencies and broken repository links.

----

Reviewer nffD raised substantive concerns regarding the formal definition of instance-level heterogeneity and causal diversity, the strength and realism of Assumption 3.5 for guaranteeing OOD generalization, and the lack of quantitative analysis of computational efficiency with respect to the number of experts. In the rebuttal, the authors clarified the concepts of instance-level heterogeneity and causal diversity, provided a detailed explanation and discussion of Assumption 3.5 and its implications for generalization, and supplemented the paper with an analysis of computational efficiency. These responses address the reviewer’s main questions and substantially improve the conceptual clarity of the work.

**Reviewer Scores:**

The concerns raised by Reviewer 4z1L have been well addressed through additional analyses, experiments, and visualizations in the rebuttal, and the reviewer is likely to maintain a positive stance toward the paper.

----

Reviewer GS29 did not increase the score during the rebuttal phase, the responses meaningfully addressed the core concerns and improved the overall clarity and completeness of the work.

----

The responses address most of the Reviewer q7vv’s concerns; however, the motivation and justification for the use of the term “OOD” could have been articulated more clearly, and the reviewer may therefore be inclined to maintain their original score.

---

These responses address the Reviewer nffD’s main questions and substantially improve the conceptual clarity of the work.

---

### Decision · Program_Chairs · 2026-01-26

Accept (Poster)